# Modulating Effect of Diet on Alzheimer’s Disease

**DOI:** 10.3390/diseases7010012

**Published:** 2019-01-26

**Authors:** Paloma Fernández-Sanz, Daniel Ruiz-Gabarre, Vega García-Escudero

**Affiliations:** Department of Anatomy, Histology and Neuroscience, School of Medicine, Universidad Autónoma de Madrid, 28029 Madrid, Spain; paloma.fernandezs@estudiante.uam.es (P.F.-S.); daniel.ruizgabarre@estudiante.uam.es (D.R.-G.)

**Keywords:** Alzheimer’s disease, nutrients, mediterranean diet, oxidative stress, inflammation, autophagy, beta-amyloid, tau

## Abstract

As life expectancy is growing, neurodegenerative disorders, such as Alzheimer’s disease, are increasing. This disease is characterised by the accumulation of intracellular neurofibrillary tangles formed by hyperphosphorylated tau protein, senile plaques composed of an extracellular deposit of β-amyloid peptide (Aβ), and neuronal loss. This is accompanied by deficient mitochondrial function, increased oxidative stress, altered inflammatory response, and autophagy process impairment. The present study gathers scientific evidence that demonstrates that specific nutrients exert a direct effect on both Aβ production and Tau processing and their elimination by autophagy activation. Likewise, certain nutrients can modulate the inflammatory response and the oxidative stress related to the disease. However, the extent to which these effects come with beneficial clinical outcomes remains unclear. Even so, several studies have shown the benefits of the Mediterranean diet on Alzheimer’s disease, due to its richness in many of these compounds, to which can be attributed their neuroprotective properties due to the pleiotropic effect they show on the aforementioned processes. These indications highlight the potential role of adequate dietary recommendations for clinical management of both Alzheimer’s diagnosed patients and those in risk of developing it, emphasising once again the importance of diet on health.

## 1. Introduction

Accumulating scientific evidence during the last years on the possible effect of diet in neurodegenerative disorders, such as Alzheimer’s disease (AD), has motivated this review to determine which nutrients may have a beneficial or detrimental role in the pathology.

Although dietary recommendations are far from being a treatment for AD, it may be able to alleviate some of the symptoms patients display or slow down the cognitive and physical decline they suffer. The Mediterranean diet includes multiple nutrients that have been proven to be favourable and high adherence to these dietary requirements decreases the risk of developing AD or mild cognitive impairment [1,2]. Namely, the protective effect has been linked to several substances with antioxidant function, such as olive oil, wine, vegetables, fruits, vitamins, and polyphenols, that are able to diminish oxidative stress and inflammation [2,3].

We propose that diet plays an essential role in AD, since it affects Aβ production, Tau accumulation and their elimination, and modulates oxidative stress and inflammation. Thus, dietetic intervention would be necessary both in AD patients and the population at risk of developing it.

To review this aspect, we aim to collect the available evidence regarding how diet affects AD and the possible effect of specific nutrients as modulators of the processes involved in the pathology. We have appraised the benefits of the Mediterranean diet as an example of an appropriate diet for these patients and evaluated the relevance of nutritional intervention in them. In addition, we have carried out a revision of the aspects of the disease that may be modified by dietetic intervention and of the nutrients that may have modulating effects on those aspects.

## 2. Alzheimer’s Disease

### 2.1. Disease Description

Alzheimer’s Disease is a neurodegenerative disorder that accounts for the majority of cases of dementia, affecting over 35 million people worldwide [3]. It typically appears in patients over 65 years old, explaining the higher prevalence found in the elderly population [4]. In the context of a progressively ageing population, where the manifestation of age-related pathologies is increasingly frequent [5], a rise of AD’s prevalence and incidence is to be expected, with some studies suggesting up to a three-fold enlargement of these data by 2050 [3].

Some of the more usual clinical features include cognitive impairment, memory loss, language disorders, sudden changes of mood and behavior, and disorientation in time and space, hampering the patient’s ability to perform normal daily activities [6,7]. It typically lasts around 10–12 years, but there is remarkable interindividual variability [8].

AD is classified in two types, according to its onset: Familial Alzheimer’s Disease (FAD) and Sporadic Alzheimer’s Disease (SAD). FAD starts between 40 and 60 years old (early-onset AD) and follows an autosomal dominant pattern of inheritance. It shows a low prevalence, accounting for barely 5% of the cases. It has been related to mutations in the genes that encode presenilin 1 and 2 (PS1 and PS2), resulting in enhanced processing of amyloid precursor protein (APP) by β- and γ-secretases, responsible for the amyloidogenic route [9,10], or in the gene that encodes APP itself [11].

The sporadic form of the disease is also known as late-onset AD, appearing after 65 years old. It has a multi-factorial origin, involving genetic traits, as well as environmental factors. While the latter may include lifestyle, diet, socioeconomic status, educational level, or the presence of comorbidities, such as obesity or diabetes [7,12], the former refers to several genetic factors that may confer protection against certain mechanisms of the disease or increase the risk of developing it. A very well-known and extensively-researched example of this is the Apolipoprotein E ε4 (APOEε4) allele [13], which constitutes the most common genetic risk factor for late-onset AD [14]. Its corresponding protein contributes to Aβ elimination, as pointed out by the fact that the presence of this allele is related to a higher Aβ concentration in the brain [15]. It has also been linked to oxidative stress, with patients carrying this allele exhibiting a more oxidised plasma than that from patients that were non-ApoEε4 carriers. This would be a consequence of the structure of the protein encoded by this allele, since it does not contain cysteine residues, therefore, lacking free thiol groups that normally act as antioxidants [14].

The neurodegenerative process observed in AD, comprising mainly, but not exclusively, neurofibrillary tangles, senile plaques, and neuronal damage and loss, is usually present in patient’s brains before they manifest the first symptoms, with some evidence pointing to mitochondrial dysfunction in sensitive neurons as one of the earliest and more widespread features of AD [16].

### 2.2. Neuropathological Markers

AD key features are, as mentioned, the presence of neurofibrillary tangles and senile plaques and neuronal loss, resulting in cerebral atrophy.

Neurofibrillary tangles are composed of abnormal tau protein aggregates [17]. Under normal conditions, tau protein contributes to the cytoskeleton structure by interacting with tubulin in order to stabilise the microtubule network [18]. However, it may suffer different post-transcriptional modifications, such as truncation or hyperphosphorylation. Although the reasons that motivate these modifications remain elusive, there is compelling evidence that this hyperphosphorylated form is prone to aggregate, leading to the formation of neurofibrillary tangles that constitute toxic intracellular accumulations, mainly located in the hippocampus [10]. Moreover, tau malfunction produces cytoskeleton destabilisation due to microtubule collapse, prompting synaptic failure that results in a loss of communication, thus contributing to AD-mediated neurodegeneration [7].

Since mitochondrial transport depends on its interactions with microtubules, this process is hindered upon tau hyperphosphorylation, causing energy deficits in presynaptic areas that may lead to synaptic disruption, given the large amount of energy required to accomplish the exchange of information between neurons.

Senile plaques consist of extracellular deposits of β-amyloid peptide (Aβ), originating from amyloid precursor protein (APP) degradation. Such deposits are cause of inflammation and neuronal death [17]. APP is a transmembrane protein present in neurons, which can be processed following two different routes: The amyloidogenic and the non-amyloidogenic pathways [10], both mediated by secretases. β- and γ-secretases take part in the first one, while α- and γ-secretases intervene in the latter one (Figure 1).

In the non-amyloidogenic route, APP is sequentially cleaved by α-secretase and γ-secretase, giving rise to truncated peptides, Aβ_17–40/42_. However, it is worth noting that is has been described that β-secretase cleavage can ensue α-secretase, thus producing alternative shorter non-aggregative Aβ fragments [14,19]. Among them, truncated peptide Aβ_1-16_ has been found to be increased in the cerebrospinal fluid (CSF) of both familial and sporadic AD patients together with a decrease in Aβ_1–42_, suggesting a possible contribution to the disease [20,21] (Figure 1). In the amyloidogenic route, on the other hand, the sequential cleavage is carried out by β-secretase and γ-secretase, leading to whole-length Aβ peptides, responsible for the formation of the aforementioned plaques. Both routes yield an amino-terminal fragment (sAPPα for the non-amyloidogenic route or sAPPβ for the amyloidogenic route) and a carboxy-terminal one (CTF83 and CTF99, respectively). The action of γ-secretase generates the APP intracellular domain (AICD), which participates in cellular signalling. Depending on the point where γ-secretase performs the cut in the amyloidogenic route, the whole-length Aβ peptide would have a different length, with Aβ_1–40_ and Aβ_1–42_ being the dominant fragments in the brain [14] (Figure 1).

In addition, a novel APP processing mechanism has been recently described. Such a mechanism consists in the cleavage of APP between the amino acids 504 and 505, by a n-secretase, consequently, producing bigger carboxy-terminal fragments after α- or β-secretase cleavage: Aη-α and Aη-β, respectively. Notably, Aη-α comprise Aβ_1-16_ within its sequence and has been proven to be neurotoxic [22]. 

### 2.3. Oxidative Stress

Both Aβ and tau accumulation can interfere with mitochondrial function. For example, the interaction of both proteins with the voltage-dependent anion channel 1 (VDAC1) results in mitochondrial dysfunction and increased oxidative stress. They can also alter complexes on the respiratory chain, as previously mentioned [16]. In any case, this enhanced oxidative stress can lead to inactivation of a presequence protease that participates in Aβ degradation at the mitochondrial level, promoting Aβ accumulation even further, which in turn would worsen the mitochondrial dysfunction and oxidative stress, inducing apoptosis in the cell [16].

Oxidative stress is one of the major risk factors for developing a neurodegenerative disorder, for it triggers the formation of free radicals that inflict damage at the cellular level as well as the molecular level, affecting proteins, lipids, and nucleic acids. This oxidative stress is due to an imbalance between ROS or free radicals and antioxidant substances, either caused by a rise of the concentration of the former or a reduction of the levels of the latter [10].

Oxidative damage in lipids provokes its peroxidation (Figure 2). This process is critical for structures containing a high quantity of lipids, such as the cell membrane, whose permeability can be greatly modified upon lipid peroxidation, ultimately leading to cell death. The process consists of three phases: Initiation, propagation, and termination.

Initiation starts when a molecule of oxygen, a singlet oxygen, or hydroxyl radicals attack a fatty acid, pulling out a hydrogen atom and giving rise to a lipid radical (L*). This is followed by the propagation step, where a lipid radical interacts with oxygen, generating a lipid peroxyl radical that may interact with another fatty acid, yielding a lipid peroxide and another lipid radical that can repeat the cycle. Termination takes place when the lipid peroxyl radical interacts with other radicals and produces a stable species. There is controversy about the extent to which this would affect AD, since increased lipid peroxidation markers in AD patients have only been found in some studies [23].

As for proteins, oxidative damage causes structural and functional changes that lead to aberrant proteins that may increase oxidative damage even further, establishing a cyclic process. In AD, ROS oxidise Tau and Aβ. These oxidised, functionally abnormal proteins accumulate in the cytosol of neurons, creating neurofibrillary tangles and Aβ plaques, with the latter increasing ROS levels, feeding the cycle of oxidative damage [24]. In addition, some of these anomalous proteins are resistant to proteolysis and are able to inhibit proteases that break down oxidised proteins [25].

Proteins are oxidised under physiological conditions. To keep them within an adequate range, there must exist a balance between them and their repair mechanisms. However, these mechanisms lose efficacy as age increases, resulting in a higher accumulation of damaged proteins [25].

To prevent neuronal death and synaptic loss due to oxidative stress, neurons rely on the repair mechanisms of the endogenous and exogenous antioxidant systems. The first include enzymes, such as superoxide dismutase (SOD), catalase, and glutathione peroxidase (GSH peroxidase), while the exogenous system consists of some vitamins (A, C, and E), polyphenols, and certain metals, such as copper, zinc, and selenium, although it is worth remarking that zinc and selenium are not actual antioxidants, but cofactors, necessary for some enzymes to perform their function [26]. SOD possesses several metallic atoms in its active site, which are essential to convert the superoxide anion to hydrogen peroxide (H_2_O_2_). GSH peroxidase depends on selenium to coordinate glutathione oxidation to glutathione disulfide (GSSG) with the reduction of oxidised compounds, such as H_2_O_2_. In this manner, GSH peroxidase prevents oxidation of lipid peroxides or H_2_O_2_, which would otherwise be metabolised to form ROS. This enzyme acts in concert with GSH reductase, leading GSSG back to GSH in the presence of NADPH (reduced nicotinamide adenine dinucleotide phosphate) to ensure correct levels of this antioxidant metabolite [27]. As for catalase, it contains iron in its core and mediates the transformation of H_2_O_2_ from cell metabolism into water and oxygen [26].

SOD and GSH peroxidase activity is decreased in AD patients, consequently, resulting in a rise of oxidative stress and oxidised proteins’ concentration in the brain, mainly in the frontal cortex compared to the occipital cortex and to a greater extent in older patients. More recent studies showed that the highest levels of oxidised proteins are found in the hippocampus and parietal cortex [28].

Oxidative stress also affects mitochondria, promoting the expression of a β-secretase linked to Aβ formation (BACE 1) and inducing conformational changes in PS1 that increase the Aβ_42/40_ ratio. These changes are usually observed in neurons situated close to Aβ plaques [29]. Mitochondrial dysfunction can be intensified by Aβ and Tau interacting with VDAC1, as previously mentioned. In addition, PS1 mutations affect mitochondrial recycling by mitophagy-mitochondrial processing by autophagy; for presenilin is necessary for lysosomal maturation during this process [16].

Recent evidence shows that brain regions undergoing protein and lipid oxidation are rich in Aβ peptides [14]. Some studies suggest that iatrogenic administration of antioxidants may be able to alleviate Aβ-induced toxicity in the brain of AD patients and prevent oxidative stress in neurons [30], but further research is needed to endorse these findings [26].

### 2.4. Autophagy

Autophagy constitutes a catabolic mechanism that eliminates abnormal misfolded proteins and damaged organelles. This process is hindered in AD, resulting in Tau and Aβ aggregation that lead to the formation of neurofibrillary tangles and senile plaques [31].

There are three types of autophagy: macroautophagy, microautophagy, and chaperone-mediated autophagy; differing on recognition mechanisms and substrate degradation pathways. Macroautophagy is the main degradation pathway for aggregated proteins, which explains its use as one of the principal AD markers. To simplify, macroautophagy will henceforth be referred to as ‘autophagy’.

Autophagy consists of five stages: Nucleation, elongation of the phagophore, autophagosome formation, autophagosome fusion with lysosomes, and recycling or degradation of sequestered biological residues [29] (Figure 3).

The process starts with activation of the kinase complex ULK (unc-51 like autophagy activating kinase). This complex phosphorylates Beclin-1, activating it. Beclin-1 is part of a complex that initiates the nucleation of the phagophore, the initial sequestering organelle formed around the substrate that needs to be processed. Under stress conditions, cells activate a number of regulating proteins, including the mTORC1 complex, whose regulation relies on phosphorylation and dephosphorylation and therefore depends on the AMP/ATP ratio, which has been linked to the cell nutritional status. mTORC1 activation phosphorylates the ULK complex, inhibiting it and consequently blocking autophagy activation. Thus, mTORC1 inhibition induces autophagy [30,31] (1 in Figure 3).

Once the process is started, the protein, LC3-II, binds to the phagophore, enabling its elongation through lipid acquisition (2 in Figure 3). LC3 is synthesised as pro-LC3, which suffers cleavage by ATG4, yielding the active form, LC3-I. ATG7 and ATG3 catalyse the binding of LC3-I to phosphatidylethanolamine from the phagophore membrane, where it becomes LC3-II. This explains that LC3-II can be used as a correlate of the number of autophagosomal membranes [32,33].

The expanded phagophore ultimately seals, generating an autophagosome that contains the cytoplasmic material that will be degraded (3 in Figure 3).

This mature autophagosome merges with a lysosome, giving rise to an autolysosome or autophagolysosome, whose interaction depends on the acid pH of lysosomes (4 in Figure 3). The hydrolytic enzymes contained in these autolysosomes break down the sequestered material, and the resulting products (lipids, amino acids, simple sugars, and nucleotides) are released by means of permeases back to the cytosol, where they can be recycled to synthesise new biomolecules.

Autophagy is an essential pathway in neurons since ageing provokes intracellular accumulation of toxic residues and damaged organelles that jeopardise homeostasis. In addition, some proteins, like Beclin-1, ATG5, or ATG7, diminish with age, hampering autophagy and enhancing Aβ build-up. This has been linked to the neuronal dysfunction of AD patients [34].

Beclin-1 induces autophagy by regulating the activity of Vps34 and by participating in membrane recruitment to generate the autophagosome [32]. These two proteins, together with ATG14 and Vps15, constitute the phosphoinositide 3-kinase complex (PI3K) responsible for the nucleation of autophagic vesicles. A failure in this pathway, as described with age, could therefore lead to Aβ accumulation and the display of AD symptoms [31]. Studies conducted in mice have also described the formation of hyperphosphorylated Tau deposits in brain that can be lessened upon autophagy induction [35]. Moreover, a deficiency of Beclin-1 in an APP transgenic mouse model diminished neuronal autophagy, giving rise to intracellular and extracellular Aβ deposits that led to neurodegeneration [32]. Thus, autophagy and Beclin-1 induction were proposed to modulate neurodegeneration and Aβ accumulation in AD model mice.

Other researchers have demonstrated that mitophagy activation by PARK2 overexpression is able to rescue the mitochondrial deficiency found in AD patients derived fibroblasts and to restore synaptic function in the triple-transgenic mouse model of the disease (3xTg) [36,37].

### 2.5. Inflammation

Inflammation is the organism’s response against a threat, carried out by inflammatory mediators, such as cytokines (Interleukin 1, tumour necrosis factor), prostaglandins, growth factors, thromboxanes, or ROS. In AD patients, Aβ plaques and neurofibrillary tangles provoke a chronic inflammatory state, stimulating the action of these mediators. These, in turn, enhance the amyloidogenic route of APP processing, increasing Aβ_42_ levels. Aβ also activates proinflammatory cytokines as tumour necrosis factor (TNF) in glial cells [38,39], as well as some enzymes related to inflammation, such as ciclooxygenase-2 (COX-2) or inducible nitric oxide synthase (iNOS) and the nuclear factor kappa B (NF-κB). This inflammatory response as a result of Aβ accumulation has been postulated to lead to the neuronal damage observed in AD [39].

## 3. Relevance of Dietary Patterns

The main focus of this work is to collect the available evidence regarding the mechanisms involved in the pathology of AD, such as oxidative stress, autophagy, inflammation, or APP processing, and how nutrients may affect them, which may be potentially helpful considering the lack of an effective treatment for this disease.

During the last years, increasing evidence suggests that lifestyle, diet, obesity, and socioeconomic status are some of the exogenous factors that play a role on AD’s development. Thus, dietary recommendations may offer a good opportunity to modulate the impact of nutrients on the pathology, helping to prevent it in populations at risk or even slowing down its progression in the most optimistic scenario.

Diet is a perfect example of the pleiotropic effect certain measures can exert on different pathways, endorsing the adequacy of dietetic intervention on AD. However, no dietary intervention has been unequivocally proven to prevent, protect, or treat AD or cognitive decline [40]. Be that as it may, there is accumulating evidence regarding the effect of certain nutrients in different processes of AD pathogenesis and other neurodegenerative diseases, as will be discussed in this work. Despite the need for this kind of research that addresses the mechanisms by which nutrients may modulate AD, it is inarguable that their potential utility is limited, at best. Nutrients are not consumed in isolation, and the effect of supplementation with specific nutrients on AD is unclear and may be overlooking synergistic or antagonistic actions between different nutrients contained in a certain food [40,41,42]. In fact, a recent network meta-analysis points out that nutrient-based interventions have a limited impact and do not accomplish significant effects [41]. Even single foods seem not to be able to correlate with an improvement of cognitive function or memory decline [40]. Thus, examining the effects of dietary patterns may be a more suitable and potentially beneficial approach. In contrast with the lack of an effect reported for nutrients, some diets have been linked to better cognitive functioning and lower rates of development of mild cognitive impairment and AD [43]. However, it is important to emphasise that there is a notable gap in the scientific knowledge when it comes to considering dietary interventions in clinical research. The Mediterranean diet (MD) has been proposed as one of the best options to modulate these processes, with a higher adherence to the MD being associated with a reduced risk for AD and cognitive impairment [43,44,45,46].

The MD contains several of the nutrients that will be discussed later. For instance, it comprises great amounts of vegetables, legumes, and fruits, which are rich in antioxidative vitamins capable of diminishing oxidative stress. It is also characterised by a high intake of unsaturated fat from olive oil, which contains polyphenols that may help reduce Aβ aggregates and ROS from mitochondria, outlining oleuropein aglycone that promotes autophagy. Further characteristics of the MD include low amounts of red meat and moderate intake of poultry, which means low quantities of saturated fats and cholesterol, which would induce rigidity and loss of fluidity of neuronal membranes in high amounts. In addition, low cholesterol levels imply less activation of β-secretase, preventing Aβ deposition. Moreover, the MD proposes moderate consumption of ω3-rich fish, which helps diminish the synthesis of proinflammatory and prothrombotic cytokines. Belonging to the ω3 series, DHA also decreases the activity of the secretases of the amyloidogenic route and indirectly potentiates the non-amyloidogenic one. On top of these observations, the MD is related to moderate consumption of red wine, usually during meals, with a positive effect due to its polyphenols [8,14,34], while also containing ethanol, which might help preserve cerebrovascular function [47].

Although these effects are a reflection of studies using single nutrients, the trend towards their potential benefits together with the evidence of improved cognitive function linked to higher adherence to the MD constitutes a compelling argument to address the need of specific dietary interventions in patients suffering cognitive decline and AD [41,45]. In this regard, it is important to highlight the results from a very recent study by Berti and co-workers assessing the effects of the MD on AD biomarker changes on a 3-year follow-up [48]. In it, the authors reported not only that Aβ deposition and neurodegeneration increased in those patients with lower adherence to MD, but also, that progression of these pathological features occurred at higher rates. The onset of such abnormalities was estimated to take place at least 1.5 years before the start of the study. In addition, the beneficial effects of the MD were underlined by the estimate that higher adherence to this diet may result in protection against AD for up to 3.5 years. As Berti et al. noted, clinical application of the MD as a measure to prevent or treat AD is not yet justified. Notably, there is a lack of studies addressing nutrient-drug interactions related to MD, which may be especially important in AD patients, who are usually polymedicated due to other age-related health concerns, which is fairly common among this population segment.

Incidentally, it is worth emphasising that during the last years, some research has been done regarding the effect of the ketogenic diet on AD. The ketogenic diet is rich in saturated fat and low in carbohydrates. Although scientific evidence pointed out that a saturated fat–rich diet would increase the risk of developing AD, its combination with low amounts of carbohydrates has been found to be potentially beneficial. In fact, a transgenic mice model of AD fed with this diet for 43 days showed a significant reduction of Aβ levels in their brains due to a decrease of APP amyloidogenic processing [49]. However, further research is needed in humans to test if Aβ reduction correlates with improved cognitive function.

In addition, some dietary patterns may affect AD risk and development indirectly, through modulation of some AD risk factors. For instance, obesity, type 2 diabetes, or cardiovascular disease have been established to increase the risk of developing AD [8,15,39,50] and certain dietary and lifestyle interventions have been proven to prevent or alleviate the impact of these conditions in health. Thus, this improvement may translate as a diminished risk of developing AD or another type of dementia [51].

The specific mechanisms by which the MD or other dietary patterns may result in benefits for AD or other pathologies is yet to be elucidated [48]. Hereby, we review some of the most important mechanisms described for different nutrients that may help modulate AD pathogenesis, always keeping in mind that synergistic activities within foods and diets are to be expected and are most likely ultimately responsible for the observed effects. Furthermore, we include information about studies conducted in humans regarding the discussed nutrients, but it is necessary to underline that, overall, clinical research has failed to prove clear preventive or therapeutic effects of any dietary intervention for Alzheimer’s disease. This is partly due to inherent problems in translational research and partly because of the notable number of confounding factors that may exist when evaluating a dietary intervention. However, the potential utility of dietary interventions drawn from the mentioned studies [43,44,45,51] constitutes a compelling argument in favour of research focusing on dietary patterns and the mechanisms by which combined nutrients may modulate chronic diseases as AD.

## 4. Nutrients That Modulate Alzheimer’s Disease

### 4.1. Unsaturated Fats (Monounsaturated and Polyunsaturated)

Monounsaturated fatty acids (MUFAs) are lipidic biomolecules containing one unsaturated carbon bond or double bond in their structure, while polyunsaturated fatty acids (PUFAs) have more than one double bond between their carbons. PUFAs are constituted by two groups: Those belonging to the omega 3 series (ω3) and those included on the omega 6 series (ω6). Within these series, the α-linolenic (ω3) and the linoleic (ω6) acids are essential fatty acids that humans need to acquire from food intake and give rise to long-chain essential fatty acids by elongations and desaturations [38]. Arachidonic acid (AA) is synthesised from linoleic acid, whereas docosahexaenoic acid (DHA) and eicosapentaenoic acid (EPA) come from α-linolenic acid (ALA) [52]. The ratio of ω3/ω6 has implications for AD (summarized in Figure 4): Both types of fatty acids compete for the same desaturases to be incorporated into cell membranes. Thus, higher amounts of ω6 acids hamper α-linolenic conversion into EPA and DHA, resulting in decreased levels of these fatty acids. Accordingly, it enhances the synthesis of proinflammatory eicosanoids from ω6 acids, such as prostaglandins, thromboxanes, and leukotrienes. These substances exert inflammatory and vasoconstrictive functions that may increase cardiovascular risk and, therefore, the probability of suffering from a neurodegenerative disorder [53].

On the contrary, ω3 fatty acids are able to reduce inflammation by different mechanisms. They can hamper AA synthesis by competing with ω6 fatty acids and block AA conversion into proinflammatory factors via EPA-mediated COX-2 inhibition. In addition, to hinder the production of inflammatory mediators, EPA gives rise to anti-inflammatory eicosanoids, as well as resolvins (E-series) that facilitate inflammation termination [53]. In mice-derived microglia, DHA is able to decrease NO and ROS release induced by lipopolysaccharide or interferon γ, thus preventing inflammation via glial modulation [54,55]. This phenomenon could be partially explained by the fact that DHA is incorporated in the cellular membrane, hampering antigen presentation, but it also seems to reduce inflammatory molecules’ expression by inhibition of p-38 mitogen-activated protein kinase phosphorylation by MKKs and promotion of PPARγ activation, modulating glucose and lipid metabolism [38,56,57]. In addition, DHA and its active derivative neuroprotectin D1 (NPD1) have been linked to the microglial phenotype M2, related to inflammation termination through secretion of anti-inflammatory mediators, such as IL-10 and IL-4. DHA also has been proposed to mediate an increase on glial phagocytic activity, both in vitro and in vivo. Regardless of this, it remains unclear if DHA can affect the glial phenotype directly or indirectly through diminished proinflammatory mediators [38,53,55]. Since the inflammatory process constitutes one of the essential mechanisms of dementia, ω3 fatty acids may have a protective effect due to these anti-inflammatory properties [58]. In a study of 2015, J. Thomas and colleagues affirmed that some physiological changes occur in the ageing brain, such as depletion of long chain omega 3 fatty acids, and this process progresses in a steeper way in AD [59]. This is consistent with lower levels of DHA found in AD patients [15,59].

DHA plays a fundamental role in the normal growth, development, and function of a nervous system, as well as in its maintenance and the preservation of the neuronal structure [37]. DHA can be incorporated in neuronal cell membranes, where it may have a direct effect on APP processing, but also indirect effects due to alteration of membranes’ fluidity that can hinder lateral movement of proteins, preventing substrate/enzyme interactions [57]. The roles of fatty acids in AD’s pathology is less clear, although some plausible mechanisms have been described. Studies in CHME3 cells (a human microglial cell line) showed increased Aβ phagocytosis along with an anti-inflammatory profile after DHA administration [58]. In an APPsw (Tg2578) transgenic mouse model, this acid prevents lipid peroxidation and reduces Aβ accumulation both in the hippocampus and in the cortical region, due to a decrease in the activity of β- and γ-secretases, avoiding Aβ peptides’ generation [59,60,61]. This agrees with evidence showing that a DHA-rich diet for 4 weeks in 12 male C57B1/6J mice decreased both the γ-secretase and the β-secretase, showing a greater effect on the γ-secretase [62]. DHA did not alter the expression of other examined genes or proteins, such as PS1 and BACE1, confirming a direct effect on the secretases of the amyloidogenic pathway. In addition, it had an indirect influence on the non-amyloidogenic pathway by increasing the transcription of ADAM-17, a gene that participates together with α-secretase in the proteolytic cleavage of APP, preventing the production of Aβ and increasing the levels of the amino-terminal fragment, sAPP-α, and the C-terminal fragment (CTF83) of the non-amyloidogenic pathway (Figure 4). Finally, it is worth noting that DHA metabolism gives rise to active derivatives, such as neuroprostanes or NPD1, which have been proposed to be neuroprotective [63], as well as D-series resolvins, which help to induce inflammation termination [50]; though their influence on the aforementioned processes, including inflammation and APP processing, are yet to be untangled.

Most of these mechanisms were recently summarised in two reviews by Grimm et al. and Cardoso et al. [60,61]. From all this it can be concluded that DHA could be beneficial in the pathogenesis of AD mainly, but not only, by reducing inflammation and diminishing Aβ deposition in the brain. In line with this, epidemiological studies support the idea that insufficient consumption of DHA is linked to a greater risk of developing AD [62].

However, in 2006, Freund-Levi and colleagues carried out a double-blind study with patients at different stages of AD from mild to moderate, where they were supplemented with 1.72 g DHA + 0.6 g EPA per day for six months. They found only a small improvement in cognitive decline in a subgroup of patients with the mildest form of AD, with no improvement in the rest of the patients [63]. This may indicate that ω3 can play a role in the prevention of AD, but not once the neuropathological condition is advanced, as suggested by a study by Fiala et al., where ω-3 supplementation stabilised the cognitive state of patients with mild cognitive impairment, but was not able to stabilise nor improve it in AD patients [64]. Both epidemiological and clinical studies second the notion that ω-3 fatty acids may only be effective prior to the onset of the disease or when patients display mild symptoms [65,66,67].

Apart from this, other interacting factors should be considered. For instance, there is evidence suggesting DHA metabolism is especially disrupted in AD with the APOE *ε4* allele, who would then constitute a subgroup refractory to the treatment with this acid [38]. This is supported by a study by Huang et al. that associated intake of fatty fish with a reduced risk of dementia in APOE *ε4* allele non-carriers [68], although a more recent meta-analysis including this study showed inconclusive results for the protective effect of fish [69]. In contrast, APOE *ε4* carriers may benefit from high-dose ω3 supplementation in pre-dementia stages, but this intervention needs to be optimised, maybe by taking advantage of advances in brain imaging techniques [70]. Differences between APOE *ε4* carriers and non-carriers may have populational consequences, considering that the distribution of the different APOE alleles varies geographically [71]. Another factor worth mentioning is the fact that most studies supplementing these nutrients use ω3 (especially DHA) alone as a specific nutrient, which may disregard the synergistic effects of whole foods containing them [60] and dietary patterns including those foods. In fact, some evidence points out that AA:EPA and AA:DHA ratios rather than the isolated compounds would be responsible for the observed effects. For instance, increased release of the anti-inflammatory cytokines, IL-10 and TNF-α, and IL-6 and IL-8 reduction is found in 1-2:1 ratios, while the levels of these pro-inflammatory mediators is increased in 4-7:1 ratios [72]. That would help in explaining the results of a double-blind, placebo-controlled trial where individuals over 60 years old received DHA-enriched fish oil over a year, displaying an improvement in memory and attention [73].

On top of all that, other possible mechanisms may be playing a role in the effects of ω3 fatty acids on AD. In this regard, the field of epigenetics may be a good opportunity to improve our understanding of how different nutrients could affect the disease. Particularly, DHA appears to participate in histone demethylation, leading to changes in gene expression and suggesting diminished apoptosis [74], but further studies shall be carried out in order to elucidate the specific mechanisms by which this would have an influence on AD’s development [75]. A summary of the most relevant of these mechanisms is displayed in Figure 4.

### 4.2. Vitamins

As seen above, ROS accumulation in the brain may be due to a decrease in the antioxidant capacity. For this reason, the intake of fruits and vegetables that contain high quantities of vitamins, such as vitamin A, C, D, and E, or the B-complex may play a preventive role against AD development, thanks in part to their potential antioxidant action. Fruits and vegetables are composed of other beneficial nutrients, such as flavonoids, whose protective effects against neurodegenerative processes are discussed later in this paper [2]. In fact, lower levels of the fat-soluble vitamins, A, D, K, and E, and water-soluble vitamin C have been associated to cognitive decline in the elderly and AD patients [69,76,77,78,79]. As for other water-soluble vitamins, higher levels of total plasma homocysteine, a biomarker reflecting the functional status of vitamins B6, B9, and B12, has been appointed as a risk factor for dementia, cognitive decline, and AD in a recent consensus paper [80]. Yet, the benefits of vitamin supplementation remain unclear, especially considering that their effects may not be the same in patients with different genetic backgrounds [81]. Additionally, the results may be affected by other confounding factors, such as the fact that people with lower levels of vitamins are more likely to be following a less healthy lifestyle that may imply other risk factors. The effects of these vitamins on AD are summarised in Figure 5 and are discussed below.

Vitamin A is essential for the development of the nervous system in adulthood and childhood, since it protects from oxidative damage to embryonic neurons [82]. These antioxidant properties have been confirmed both in in vitro and in vivo AD models, along with antioligomeric and neuronal-protective effects [77,79]. Functions related to Aβ formation and inflammation have also been described [83]. For example, APP amyloidogenic processing seems to be hampered due to modifications on intracellular sorting of secretases by all-trans retinoic acid (ATRA) [76]. A blind study with APP/PS1 double-transgenic TgB6C3 (APPswe, PSEN1dE9) mice receiving trans retinoic acid, a vitamin A metabolite, dissolved in normal saline containing 5% of dimethylsulfoxide three times weekly by intraperitoneal injection (20 mg/kg) reported that this acid was able to reduce tau hyperphosphorylation by decreasing the activity of a CDK5 kinase involved in the abnormal phosphorylation of tau. Similar findings were reported in a study using tamibarotene, a synthetic retinoic acid analogue that upregulates ADAM10, an enzyme involved in non-amyloidogenic APP processing [84]. Also, it restricts Aβ deposition by hampering the production of C-terminal fragments from the amyloidogenic processing of APP. These changes entailed a cognitive improvement in these mice. As for inflammation, this same study reported decreased activation of microglia and astrocytes [85]. Moreover, vitamin A and its derivatives have demonstrated a transcription-regulating effect on several genes related to AD in vitro and in vivo, such as ADAM9 and ADAM10, APP, or BACE, among others [76,86,87,88]; as well as a potential role in reducing histone acetylation through its antioxidant properties, which may be also attributable to vitamins C and E [74].

Nevertheless, however promising these effects may seem, it is important to consider the safety profile of retinoids’ administration. For instance, ATRA is toxic at its chemotherapeutic doses (45 mg/m^2^/day) and the prolonged use of retinoids may result in gastro-intestinal haemorrhage [79], which may limit, to say the least, its therapeutic utility until toxicity assays and clinical trials are carried out.

Vitamin E and its active form, α-tocopherol, are soluble in lipids and have a considerable antioxidant potential, with the latter being absorbed and accumulated in humans, while vitamin C or ascorbic acid is soluble in water and prevents the oxidation of low-density lipoproteins. The literature regarding their effects on AD is also inconclusive. Vitamin C exerts an antioxidant action that may be helpful for AD, but also antioligomeric properties have been attributed to it in APP transgenic mice [89]. Additionally, the administration of high-dose vitamin C in Gulo KO mice with a 5XFAD mice background (a cross-bred between an AD mice model and mice unable to produce vitamin C) resulted in reduced amyloid plaques in hippocampal and cortical regions; and short-term high-dose infusion enhanced spatial learning and memory in APP/PSEN1 and wild-type mice [90]. In line with this, some clinical studies suggest that supplementation with ascorbic acid may decrease the risk of AD, with epidemiological studies supporting lower plasma concentrations of vitamin C among AD patients [79]. However, a considerable body of research found no association between vitamin C intake and decreased risk for AD, some of them pointing out a lack of effect on plasma Aβ levels [79,91].

As for vitamin E, it is actually a family of four tocopherols and four tocotrienols, some of which have shown antioxidative and anti-inflammatory activities [92]. Apart from these properties, vitamin E has been reported to prevent Aβ-induced tau phosphorylation in vitro and in APP/PS1 mice, by inhibiting p38MAPK phosphorylation [93,94]. Moreover, antiamyloidogenic activity and regulation of secretases mRNA has been attributed to vitamin E [79]. Clinical studies seem to be more conclusive than those for ascorbic acid, with an inverse correlation between vitamin E levels and AD’s progression rate, and vitamin E intake showing a reduction in cognitive decline, as reviewed by Visioli et al. [79], and improving dependence and survival of these patients [95]. Interestingly, some studies suggest rather a synergistic activity between vitamins C and E, since ascorbic acid reduces the oxidised α-tocopherol, regenerating it [96], and vitamin E can in turn restore oxidised vitamin C at the membrane-cytoplasmic interface [66], although the meta-analysis performed by Cao and co-workers [69] points out that the association between total antioxidant intake and lower AD risk may be attributed to vitamin E intake, but not to vitamin C nor flavonoids. However compelling these results may be, it is worth remarking that systemic antioxidant actions of vitamins C and E in humans have not been indisputably proven and that the majority of trials reporting vitamin E effects used α-tocopherol supplementation [79], potentially disregarding the aforementioned mechanisms of other vitamin E forms [92]. In fact, some studies suggest that these other forms may exert the most neuroprotective results and that high concentrations of α-tocopherol in the brain may increase AD pathology [74].

Research about the effects of vitamins K and D on AD is far more infrequent. Lower vitamin K levels have been found in plasma samples of APOE ε4 carriers compared with healthy subjects with other APOE alleles [97]; but studies on AD patients are lacking and a number of confounding factors may be involved in these results, such as lower intake of vitamin K in the diet of AD patients, as reported by Presse et al. [78]. Nonetheless, it should be mentioned that vitamin K deficiency has been linked to a loss of vasomotion and increased risk of haemorrhage [79], which might increase the risk for dementia due to cardiovascular disease. Also, vitamin K analogues have been reported to inhibit Aβ aggregation and protect against its cellular toxicity in silico and in vitro [98], but deeper research should be done to validate these results for vitamin K or its utility in AD animal models and patients. More studies have been carried out with vitamin D, most of them showing a correlation between low vitamin D levels and increased risk of dementia and linking it to AD pathology in vitro and in vivo, via increased Aβ clearance and phagocytosis that led to diminished accumulation [79,99]. Also, anti-inflammatory and antioxidative properties have been described for vitamin D: The former in mice models, probably through proinflammatory cytokines’ suppression, and the latter in vitro by exerting neuroprotective effects against free radicals and inhibiting iNOS [99]. Still, clinical trials have failed to reach a conclusion about vitamin D effects on AD, with high inter-study heterogeneity [79].

Finally, with regards to the B-complex group of vitamins, B12, B6, and B9 or folic acid should be mentioned. They have been proposed to have a beneficial effect on AD due to their influence on the metabolic cycle of homocysteine, a sulphur amino acid derived from methionine metabolism (Figure 5). In this metabolic pathway, homocysteine becomes S-adenosylmethionine (SAM), which is a donor of methyl groups in other metabolic reactions, including DNA methylation. This process involves a series of cofactors, which are folic acid (vitamin B9), vitamin B12, and vitamin B6. Folic acid and vitamin B12 act in coordination to transform homocysteine into methionine, while vitamin B6 converts homocysteine into cysteine. Folic acid from the diet is transformed into tetrahydrofolate (THF) that is in turn converted into 5,10-methyltetrahydrofolate (5-10MTHF). This compound gives rise to the active form of folic acid: 5-methyltetrahydrofolate, which is the methyl group donor compound that allows methionine production and, consequently, DNA methylation [100].

Therefore, a deficit of these cofactors involved in the homocysteine pathway blocks the cycle, increasing homocysteine levels and hampering DNA methylation. DNA hypomethylation activates the γ- and β-secretases of the amyloidogenic pathway, promoting Aβ production [100]. Other epigenetic mechanisms have been proposed, such as B12 deficiency inducing presenilin also due to DNA hypomethylation. In addition, folic acid seems to be associated to DNA hypomethylation on a reverse U base instead of a linear correlation, since excessive folic acid can cause accumulation that leads to abnormal B9 metabolism, resulting in DNA hypomethylation, as would lower B9 levels [74]. Related to this, it is important to consider that certain polymorphisms are responsible for an enhanced susceptibility to alterations in these metabolic routes, leading to insufficient DNA repair and methylation and higher levels of homocysteine, among other consequences [101]. On the other hand, if homocysteine accumulation reaches concentrations higher than 15 µmol/L, it is considered hyperhomocysteinemia. This situation produces both a greater cognitive impairment and an increase of vascular dysfunction associated with dementia, due to atherogenic and prothrombotic properties, thus constituting a risk factor for cardiovascular disease with possible relevance in AD [8,15]. To evaluate this hypothesis, a study was conducted in AD patients with low levels of folic acid and vitamin B12, but elevated homocysteine. The levels of homocysteine were determined by high-performance liquid chromatography with fluorescence detection [102]. The increase of homocysteine in some cases produced microinfarcts, which were related to Aβ deposition and neurofibrillary tangles, thus linking it to the pathology of dementia.

Some authors propose vitamin B12, B9, or B6 supplementation as a means to reduce homocysteine levels, based on the role they play in its metabolism [8,103]. This accords with previous results showing that female transgenic mice model of AD Tg2576 overexpressing the human APP with the Swedish mutation (K670N/M671L) that were fed a diet low in folic acid (<0.2 mg/kg), vitamin B12 (<0.001 mg/kg), and vitamin B6 (<0.1 mg/kg) displayed hyperhomocysteinemia, which caused an increase in Aβ as a consequence of enhanced γ-secretase activity [104].

It is worth noting that, despite contradictory publications on the role of homocysteine in humans, there is a recent consensus paper by Smith et al. gathering clinical evidence about the relevance of homocysteine in dementia [80]. In this paper, the authors concluded that the experimental evidence is more than enough to attribute a causal role to elevated plasma total homocysteine (tHcy) in cognitive impairment and dementia. Conversely, the administration of B6, B9, and B12 vitamins led to lower tHcy and this slows brain atrophy and cognitive decline. Thus, they state that public health measures should be applied, such as screening for tHcy in the population at risk and B vitamin supplementation should be offered to those with higher levels of tHcy, though an adequate threshold for supplementation is still to be determined.

### 4.3. Polyphenols

Polyphenols are antioxidant substances with a potential neuroprotective function against AD since they seem to decrease Aβ levels [3]. They are obtained from the secondary metabolism of plants and can be found in the skin, pulp, or seeds of some foods, such as grapes. They are characterised by the presence of an aromatic ring and one or more hydroxyls groups (OH) in their structure. These compounds may have a role in the prevention and treatment of neurodegenerative diseases due to their antioxidative and anti-inflammatory properties [105], but also by means of other less known mechanisms, such as their capability of modulating intracellular signal pathways and gene expression [66] or mitochondria-enhancing actions [79]. The mechanisms by which different polyphenols may modulate AD are summarised in Figure 2 and Figure 6.

A polyphenols-rich food is the extra virgin olive oil used as the main source of monounsaturated fats in the Mediterranean diet. One of the most characteristic polyphenol in extra virgin olive oil is oleuropein aglycone (OLE), which induces autophagy thanks to its intervention in different routes [34] (Figure 6A). The activation of OLE-mediated autophagy is related to increased expression of certain autophagic markers, such as Beclin-1, cathepsin B, p62, and LC3, both in vivo and in vitro. In addition, OLE inhibits mTORC through the activation of Ca^2+^ calmodulin kinase/AMPK mediated by the increase in intracellular Ca^2+^. As previously stated, mTORC is an inhibitor of autophagy, so its inactivation induces this process. In addition, OLE also mediates the activation of autophagy genes by activating sirtuins, such as SIRT1 and histones deacetylation. SIRT1 activates many transcription factors via deacetylation, such as NF-κB, p53, and FOXO, which promote autophagy. These transcription factors, in turn, may activate autophagy genes. SIRT1 also deacetylates and directly activates autophagy proteins, such as Atg5, Atg7, and Atg8 [106,107]. 

However, it is important to emphasise that polyphenols are a very broad group that include various compounds, such as catechin, epicatechin, quercetin, resveratrol, curcumin, etc.

Catechin and epicatechin are found in green tea and belong to the group of flavanols whose neuroprotective capacity is associated with its antioxidant activity [108]. Epigallocatechin gallate (EGCG), the main active component of green tea, is able to increase the activity of catalase and SOD, thus decreasing the oxidative stress of neurons, protecting them from the cytotoxic effects of Aβ [10] (See Figure 2). It also exerts anti-inflammatory actions by downregulating the expression of pro-inflammatory molecules, such as TNF, IL-1 and TGB, although high doses stimulate expression of TNF and IL-6, exacerbating inflammation [109] (Figure 6B), which calls for further research to elucidate the mechanisms by which this may occur. In addition, metal chelation and free radical sequestering activities have been reported, along with modulation of genes with anti-apoptotic action and several signalling pathways, including MAPK, PKA, and PI3K, as reviewed by Singh et al. [110] (Figure 2).

Quercetin belongs to the group of flavonols and possesses neuroprotective properties due to its participation in the Nrf2 pathway, which is an important factor against oxidative stress since it promotes the transcription of genes that encode antioxidant enzymes, such as SOD (Figure 2). It can also block neuroinflammatory factors, such as NF-κB, and induces autophagy to eliminate damaged material from the inside of autophagosomes by activating SIRT1, as previously mentioned [33,109,111] (Figure 6B,C). In addition, quercetin appears to be able to enhance mitochondrial biogenesis, thus helping prevent neuronal degeneration (Figure 6A) [109].

Resveratrol is a phenolic compound belonging to the family of stilbenes. This compound affects different mechanisms related to the pathology of AD, such as oxidative stress, inflammation, or APP processing (Figure 2 and Figure 6). It diminishes oxidative stress by reducing the synthesis of nitric oxide and increasing the levels of glutathione (Figure 2). It also reduces inflammation by inhibiting the transcription of cytokines, such as TNF or the enzyme cyclooxygenase-2 (COX-2), and supressing the activation of glial cells (Figure 6B,C) [108,109]. Besides, it can prevent the formation of Aβ aggregates by inhibiting β-secretase activity [110] and destabilising plaques that were already formed (Figure 6B) [108]. As pointed, resveratrol also exerts widely known antioxidant actions. This is consistent with a study showing that treatment with resveratrol reduces malondialdehyde (MDA) peroxidation in an AD adult male Sprague-Dawley rats model weighing 500–550 g [111], MDA being a product derived from lipid metabolism that is used as a method to determine lipid peroxidation in the hippocampus of these animals (Figure 2). To investigate the effect of resveratrol on the Aβ-induced neurotoxicity, resveratrol (100 µM) was injected every day 30 minutes after the Aβ_25–35_ administration. The results showed the ability of resveratrol to reverse the Aβ-mediated overproduction of MDA and iNOS activation, reducing oxidative stress. Moreover, resveratrol has been reported to have a neuroprotective activity mediated by activation of SIRT1 that results in deacetylation of some substrates, such as p53, whose acetylated form induces apoptosis and senescence [108]. Although the benefits derived from it are arguable, with some authors proposing a bimodal role of SIRT1 depending on the extent of its activation (for example, excessive inhibition of p53 may hinder its tumour-repressor activity, potentially promoting carcinogenesis) [107,112,113]. Reasonably, the effects of SIRT1 activation on autophagy pathways discussed for other nutrients would also be applicable to resveratrol [114]. Other cellular mechanisms involving PPAR activation and the PI3K/Nrf/keap pathway have been described [115] and are summarised in a very recent review by Velmuragan et al. [108]. However, as published by Lange and Li [116], clinical trials testing resveratrol as a potential treatment for neurodegenerative diseases have been inconclusive, even if the administration of resveratrol-rich foods, like grapes, seem to have improved effects with respect to the administration of resveratrol itself. This may be due to the synergistic effect of different phytochemicals found in grapes, but is important to consider that a lack of efficacy in humans might occur because of differences in metabolism between mice and humans or other confounding variables, such as concomitant medication (very frequent in the elderly) and the lack of reliable biomarkers for antioxidants [117]. 

Finally, is also worth mentioning curcumin, a phenolic antioxidant from turmeric, that has numerous beneficial effects. Its antioxidative power is due to its bioactive compounds (curcuminoids), which are able to sequester free radicals, protecting the brain against lipid peroxidation [118,119] (Figure 2). To check the ability of curcumin to inhibit both inflammatory and oxidative damage, a study with a transgenic APPsw mice model of AD (Tg2576) was carried out [118]. Mice were treated with high (5000 parts per million) and low (160 ppm) doses of curcumin for six months before being euthanised and untreated ones were fed containing no drug. Treated mice had lower levels of oxidised proteins compared with untreated ones, with lower amyloid loads and smaller senile plaques. Accordingly, this substance reduces both the inflammation and the oxidative damage caused by the presence of peptide Aβ. Curcumin also has direct anti-inflammatory activity, for it can inhibit the transcription of the nuclear factor NF-κB (Figure 6B). It can also reduce oxidative stress by inhibiting the nitric oxide synthase responsible for the synthesis of nitric oxide (Figure 2) as well as by inhibiting lipoxygenases and COX-2, whose synthesis would produce prostaglandins and proinflammatory leukotrienes (Figure 6C). Also, low-doses of curcumin showed a 39% reduction of insoluble amyloid fragments and a 43% reduction of plaque burden in the brains of treated mice (Figure 6B) [118,119]. Additionally, curcumin has been reported to act as a chelator of some metals, such as Fe^2+^ and Cu^2+^, that participate in the formation of free radicals and increase Aβ aggregation [120,121] (Figure 2). In spite of this, the authors affirmed that one of the main issues of curcumin is its poor water solubility, which may explain its low bioavailability following oral administration or through the parenteral route and would constitute one of the causes of its failure in randomised control trials for AD [121]. Other roles of curcumin that may be of importance in AD, including reduction of TNFα and caspases, rise in BDNF, and interaction with different signalling pathways, have been described and are also summarised by Velmurugan et al. [108].

Importantly, these nutrients seldom appear isolated on single foods, but rather they are found in combinations, with different relative concentrations. Because of this, several studies reported the effects of polyphenols-rich foods [79] or antioxidants as a group containing polyphenols and vitamins [69] on AD, instead of the specific nutrient that may cause those effects. Thus, green and black tea, cocoa, coffee, turmeric, or wine have been reviewed by Visioli and Burgos-Ramos [79]. As an example, green tea contains antioxidant, anti-inflammatory, and neuroprotective compounds, including flavonoids, catechins, and caffeine, and it is more likely that their effects are the result of their interaction. Additionally, they also interact with genetic and environmental factors, which may explain the different results obtained when evaluating the association between green tea intake and AD in different subsets of population. Namely, cohort studies in Asia showed diminished cognitive decline with higher green tea intake [122], while a recent prospective study carried out in Germany reported a lack of this effect [40], maybe due to differences in the metabolism or purity of these compounds within green tea.

On top of the aforementioned evidence, epigenetics mechanisms have been described for several polyphenols. For instance, resveratrol activates SIRT1 and ADAM10 (Figure 6B) and displays effects similar to caloric restriction, maybe by inhibiting SIRT3, and curcumin inhibits PSEN1 activation, resulting in reduced Aβ production [74] (Figure 6B).

In any case, it is important to remark that poor bioavailability and internal metabolism of polyphenols is a major concern when it comes to its clinical application [123]. Thus, deeper research is needed to contrast their effects in humans and to which extent they may make a real difference in the development of AD, as well as to improve these issues through pharmaceutical formulation to assure treatment efficiency.

### 4.4. Moderate Alcohol Consumption

The noxious effects of ethanol have been widely proven. Regarding its effect on neurologic disorders, it has been confirmed that large amounts of alcohol promote cognitive impairment, producing an increase in the release of acetylcholine in the hippocampus that may result in a loss of memory and attention [2]. In addition, neuroinflammatory effects of ethanol both central and peripheric have been described [124]. On the one hand, direct mechanisms via toll-like receptors’ (TLR) activation have been proposed from data obtained in a rodent model of long-term alcohol dependence, where knockout mice for TLR4 did not show an increase of inflammation in comparison with control mice [125] (Figure 7A). On the other hand, indirect microglial activation has been identified in clinical studies in post-mortem samples of alcohol-dependent patients [126], presumably due to increased pro-inflammatory cytokines, which are known to influence the microglial phenotype [127]. These mechanisms would act additively, enhancing Aβ pathology in AD patients [124]. Preclinical studies with AD mouse models showing thiamine deficiency as a result of alcohol abuse also evidence exacerbated amyloid plaques, along with increased APP and BACE1 levels, which would support this hypothesis [128,129]. Moreover, recent findings suggest that alcohol consumption may have epigenetic effects by influencing S-adenosylmethionine metabolism, which could lead to DNA hypomethylation or histone demethylation upon chronic consumption [74].

However, it is important to consider that these studies evaluated severe alcohol consumption. In contrast, several epidemiological studies point out the potential beneficial role of a moderate consumption of alcohol by reducing dementia risk [74], although great variability has been reported, probably arising from a combination of confounding factors, such as population and individual differences, drinking patterns, sex, or interaction with medication or other lifestyle habits [64,125]. For instance, a moderate consumption of red wine has been associated with a lower incidence of AD among men, but increased risk for AD in women [37]. The beneficial effects could be explained partly by other components found in red wine, including antioxidants, like polyphenols, namely a complex mixture of flavonoids, such as anthocyanins and flavanols, and non-flavonoids, such as resveratrol and gallic acid, which may prevent the oxidative damage manifesting in dementia [2,37]. Although it is likely that ethanol itself plays a major role, since risk reduction has been described across different alcoholic beverages, as stated in a consensus document published by several experts [125]. Additionally, coronary blood flow increase upon red wine intake has been reported to be practically identical to that produced by the consumption of an equivalent volume of ethanol [44]. Regarding the effect of red wine on women, it may be that they are more susceptible to deleterious effects of alcohol, as the same study reported a more rapid memory decline with white wine intake in women with respect to men [37].

As for the mechanisms by which moderate alcohol consumption may protect against cognitive impairment, there are still some gaps in our knowledge. It has been demonstrated that ethanol exerts a cardioprotective effect by increasing myocardial blood flow, which can help reduce the risk of cardiovascular events [47,130] (Figure 7B). The vasodilating action of ethanol is due to the activation of nitric oxide synthase, and its ability to stimulate the transient receptor potential vanilloid type 1, a channel present in nerve terminals that activates the calcitonin gene related peptide (CGRP). It is precisely this CGRP, the vasodilator peptide, that leads to increased coronary blood flow. Since cerebral circulatory disorders predispose to cognitive dysfunction, the preventive effect of ethanol on cerebrovascular risk could protect against dementias, like AD [47]. Additionally, cohort studies showed that moderate consumption of ethanol produces a decrease in inflammatory mediators, such as interleukin 6, fribrinogen, and tumour necrosis factor, and thus exert an anti-inflammatory action [47,130]. Lastly, a study on cultured hippocampal and cortical neurons treated with 0.02–0.08% concentrations of ethanol proved reduced synaptic damage induced by Aβ and synuclein [131].

Nonetheless, to analyse the possible protective effect of moderate alcohol consumption, more studies are needed, with specifications of the type of alcohol employed being necessary, as qualitative and quantitative composition vary between different alcoholic beverages and synergistic or antagonistic activities cannot be ruled out [2]. For example, resveratrol and ethanol present in red wine exert opposite actions on the enzyme, nitric oxide synthase, with the former inhibiting it and the latter activating it [47,111] (Figure 2 and Figure 7). Thus, it is important to denote that there is not enough scientific evidence to recommend a moderate consumption of alcohol to abstemious people as a method to prevent cognitive impairment, especially given alcohol’s potential detrimental effects and the wide variability found among subjects and studies [130].

### 4.5. Trehalose

Trehalose (TRH) is a disaccharide present in yeast, insects, plants, and bacteria. The mechanisms by which it may be helpful in AD pathology remain elusive, but some relevant results have been described (Figure 8). For instance, TRH has been proven to inhibit Aβ aggregation and can rescue the phenotype of transgenic APP/PS1 mice of C57BL/6J eight-month old and wild-type littermates [132,133]. Also, TRH is a potent inducer of autophagy, since it enhances the generation of autophagosomes, as reflected in an increase of LC3-II, by acting on intracellular targets still unknown, but independent of mTOR inhibition. Related to this, TRH has been reported to increase progranulin (PGRN), a secreted growth factor that regulates lysosome homeostasis, mainly by upregulating the expression of its gene, GRN [134], resulting in enhanced autophagy flux. These studies were performed in patient-derived neurons and in an in vivo mouse model with GRN haploinsufficiency. Importantly, PGRN deficiency has been linked to different neurodegenerative disorders, including AD, while its overexpression has been demonstrated to reduce Aβ toxicity and accumulation in an AD mouse model [134,135].

Additionally, this trehalose-mediated autophagy activation clears lipofuscin, a lysosomal storage material found in AD patients. Another point of Aβ pathology worth mentioning is that TRH seems to alter APP trafficking and metabolism [136]. Specifically, upon trehalose treatment, γ-secretase cleavage is prevented, without inhibiting the enzyme. Thus, it is to be concluded that CTFs are redistributed, decreasing the amount that reach endolysosomal compartments, where γ-secretase mainly resides [136,137]. Regarding other aspects of AD pathology, this disaccharide increases tau protein uptake from the medium, accelerating its degradation by lysosomes, thus decreasing tau aggregates in a murine model of AD [133,138]. Moreover, antioxidant and anti-inflammatory properties have been described for TRH in vitro and in vivo, which may potentially contribute to alleviate the pathogenesis of AD attributed to these processes [139,140]. Finally, TRH can act as a chemical chaperone, avoiding the formation of protein aggregates and enhancing its degradation, hence its neuroprotective effect in tauopathies as AD [134,141], albeit diminished aggregation via ubiquitin-proteasome system upregulation has not been ruled out [136]. These mechanisms are presented in Figure 8.

### 4.6. Cholesterol, Saturated Fats, Trans, or Hydrogenated Fats

Cholesterol is a lipid that constitutes an essential component of cell membranes, lipoproteins, and steroid hormones. Several authors confirm that a diet rich in saturated fats (SFA) and low in unsaturated fats produces a decrease of high-density lipoprotein levels (HDL) and an increase of low-density lipoprotein (LDL) by reducing LDL receptor expression in the liver. This has some influence on the pathology of AD, both directly and through indirect increased risk of cardiovascular disease [2,53,66,142].

A study was carried out in a hemizygous double mutant PS1/APP mice model of AD where the animals received orally a daily dose of 250 mg/kg of BM15.766 [(4-(2-[1-(4-chlorocinnamyl)piperazine-4-yl]ethyl]benzoic acid)], a drug that reduces cholesterol levels by inhibiting the enzyme, 7-dehydrocholesterol-D7-reductase, that catalyses the last step of cholesterol biosynthesis. The mice model was obtained by cross-breeding a hemizygous transgenic mice expressing familial AD mutant, human APP _K670N, M671L_ (line Tg2576) and a homozygous line (line 8.9) of mice expressing the human familial, mutant PS1_M146V_ [143]. They observed a reduction of the average values of both Aβ_1–40_ and Aβ_1–42_ in the mice brains. In addition, this induced hypocholesterolemia increased the cleavage products of α- and γ-secretases, enhancing the non-amyloidogenic pathway, so it is suggested that the reduction of Aβ peptides in the mouse brain is due to these changes in APP processing. These results agree with other cross-sectional analysis comparing the prevalence of AD in three type of patients 60 years or older, subdivided by their usual medication: (1) Entire population, (2) patients receiving statins (inhibitors of HMG-CoA reductase, an enzyme involved in cholesterol synthesis), or (3) patients receiving other medication to reduce hypertension or cardiovascular issues. In this study, Wolozin et al. observed that patients receiving statins (lovastatin or pravastatin) had a 69.6% reduction in AD prevalence. Interestingly, only patients treated with statins showed a lower prevalence of this type of dementia compared to patients who used other medication to treat hypertension or cardiovascular disease; but a limitation of this analysis is that the exact dose of drugs administered to these patients is not reported [144].

Several clinical studies on the role of cholesterol in dementia are available. Anstey et al. concluded in a recent meta-analysis that high levels of total plasma cholesterol (>6.5 mmol/L) in mid-life may constitute a risk factor for AD later in life [145]. Supporting this idea, Wang et al. have reported increased cholesterol levels in patients with mild cognitive impairment [146]. Nevertheless, studies considering different subsets of lipoproteins associated to cholesterol and different population groups are needed to ratify these findings [145]. In this respect, studies addressing a possible relation between HDL levels and dementia have been discrepant, but a recent review by Koch and Jensen [147] points out that moderate to large sample size studies suggest an inverse correlation between HDL levels and the risk of AD.

Future long-term data on the use of statins as a cholesterol-lowering strategy and the impact that this may have on dementia development may shed some light about the importance of cholesterol on AD development [145]. However, the first statins were commercialised in the late 1980’s [148] and the gathering of these data would need a two to three decade follow-up, considering cholesterol plasma levels [145], so getting these data in a short-term period would come as a surprise. Also, it is worth emphasising that cholesterol uptake from the circulation to the brain is prevented by the blood-brain barrier, and thus, cholesterol metabolites, such as 24-hydroxycholesterol and 27-hydroxycholesterol, may be more appropriate surrogates of cholesterol levels in the brain [146].

Of note, apolipoprotein E plays a fundamental role in cholesterol metabolism and has been extensively linked to cardiovascular disease and dementia, with the APOE ε4 allele proposed to be a major risk factor for developing AD [149,150,151]. In fact, APOE ε4 carriers have increased amyloid plaques and phosphorylated tau in their brains and may be involved in other processes, such as mitochondrial dysfunction or impairment of mitochondrial motility; although the specific mechanisms are still under discussion [150] and may constitute a link between cholesterol levels and AD independent of cardiovascular disease. In line with this, abnormal cholesterol metabolism is stated as a key feature in AD. Indeed, the levels of the aforementioned cholesterol metabolites in CSF have been proposed as potential AD biomarkers in combination with Aβ42, total tau, or phospho-tau [146], with 24-hydroxycholesterol being associated to increased APOE in astrocytes, as well as Aβ and tau levels in AD subjects, while 27-hydroxycholesterol has been linked to hypercholesterolemia and AD [152].

Lastly, due to the lipidic nature of cholesterol, it is expected that it may undergo the same modifications that other lipids, including oxidation, a potentially relevant mechanism regarding AD pathology, as previously discussed [23,24]. Oxidised cholesterol might play a role in senile plaques’ accumulation and inflammation through macrophage recruitment and complement activation [153].

As for saturated and trans fats, the available evidence points towards an increased risk of developing AD with increased dietary intake of saturated fats; but studies are inconclusive regarding trans fats intake and AD risk [66]. In a review by Barnard and co-workers, the intake of saturated fats was positively correlated to AD and cognitive decline in the majority of studies [154]. Studies with trans fatty acids are more infrequent and this review reported mixed results.

### 4.7. Food Products Containing Alzheimer’s Disease Modulating Nutrients

Finally, it is important to consider the food products that contain the nutrients that have been discussed, if one aims to include them in their diet. Table 1 provides an overview of nutrient-rich foods for each of the proposed nutrients [155,156]. Finally, it cannot be overemphasised that changes in dietary patterns might be responsible for the potential beneficial outcomes of dietary intervention, rather than single-nutrient or even single-food supplementation.

## 5. Conclusions

The present review gathers the scientific evidence existing up-to-date showing the potential modulating effect of diet on Alzheimer’s disease, due to the actions of nutrients on different processes, such as Aβ accumulation, tau hyperphosphorylation, oxidative stress, inflammation, or autophagy. Altogether, this evidence brings to light the relevant role that nutritional intervention might play in these patients and in the population at risk of developing AD, enlightening once again the importance of dietary patterns in health maintenance. Further research is needed to clarify whether early intake of food containing these nutrients or, more likely, adherence to a certain dietary pattern including such foods would be useful as a preventive method for AD. It is clear that nutrients have a direct or indirect effect on the processes that lead to the neurodegeneration observed in AD, but the extent to which these effects may translate into actual modulation of the disease remains unclear. Thus, it is crucial to analyse the effects of dietary intervention in the long term in these populations, to which epidemiologic longitudinal studies should be carried out, especially considering that AD starts years before the first symptoms appear. Additionally, it is essential to focus future research on estimating the amount of each nutrient that is required to exert the beneficial action, especially referring these to Dietary Reference Intakes, along with information about toxic doses, food-drug interactions, and ideal synergistic effects arising from defined combinations that may result in a diet specifically optimised for AD.

## Figures and Tables

**Figure 1 diseases-07-00012-f001:**
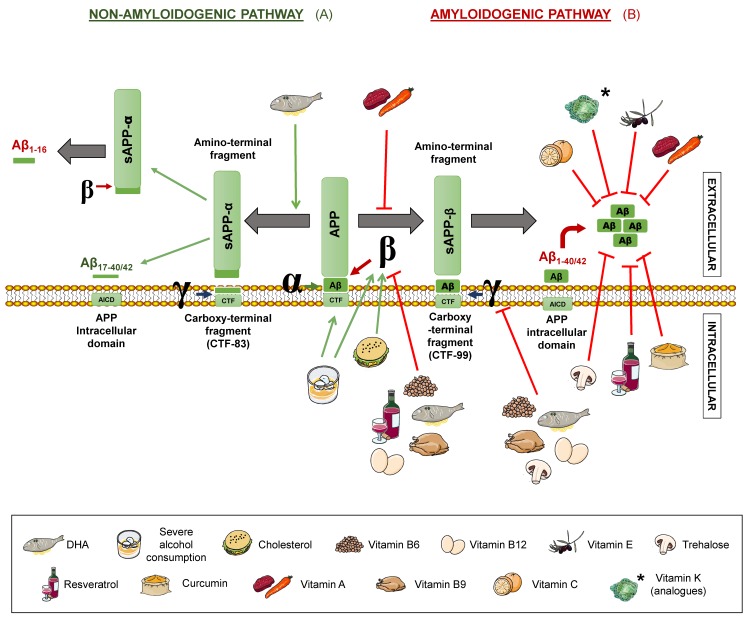
Amyloid precursor protein (APP) processing pathways and its modulation by different nutrients. The non-amyloidogenic pathway (**A**) occurs upon sequential cleavage by α- and γ-secretases (non-pathological situation) or by α- and β-secretases (pathological situation), while the amyloidogenic route (**B**) occurs when cleavage is carried out sequentially by β- and γ-secretases. Letters α, β, and γ represent each secretase. APP: amyloid precursor protein. sAPPα: soluble α-APP. sAPPβ: soluble β-APP. Nutrients that are proposed to interfere with APP processing are shown in the legend. Green arrows represent activation. Red truncated arrows show inhibition.

**Figure 2 diseases-07-00012-f002:**
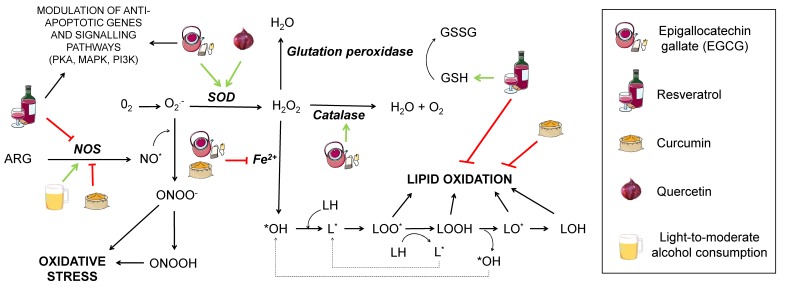
Effect of different nutrients on oxidative stress generation. Several pathways contributing to oxidative stress are shown interconnected. ARG: arginine. NOS: nitric oxide synthase. SOD: superoxide dismutase. NO*: nitric oxide. ONOO^−^: peroxynitrite anion. ONOOH: nitric acid. O_2_^−^: superoxide. H_2_O_2_: hydrogen peroxide. GSH: reduced glutation. GSSG: oxidised glutation. *OH: hydroxyl radical. LH: lipid. LOH: lipid alcohol. L*: lipid radical. LOO*: radical peroxidised lipid. LOOH: peroxidised lipid. LO*: alkoxyl. Nutrients that are proposed to interfere with the mentioned oxidative stress pathways are shown in the legend. Green arrows represent activation. Red truncated arrows show inhibition.

**Figure 3 diseases-07-00012-f003:**
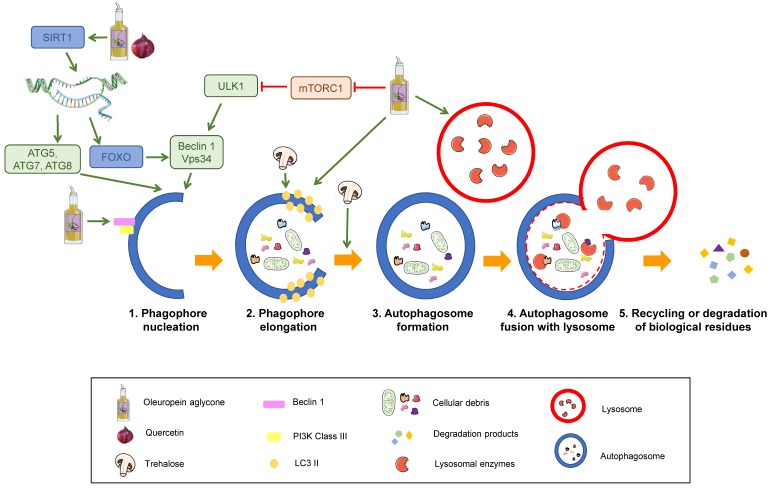
Stages of the autophagy process and its modulation by nutrients. Sequential steps of degradation by autophagy. SIRT1: Sirtuin 1. ATG: Autophagy-related protein. ULK1: unc-51 like autophagy activating kinase 1. FOXO: Forkhead box class “O” proteins. VPS34: Vacuolar protein sorting 34. mTORC1: mammalian target of rapamycin complex 1. Nutrients that are proposed to modulate autophagy are displayed in the legend. Green arrows represent activation. Red truncated arrows represent inhibition.

**Figure 4 diseases-07-00012-f004:**
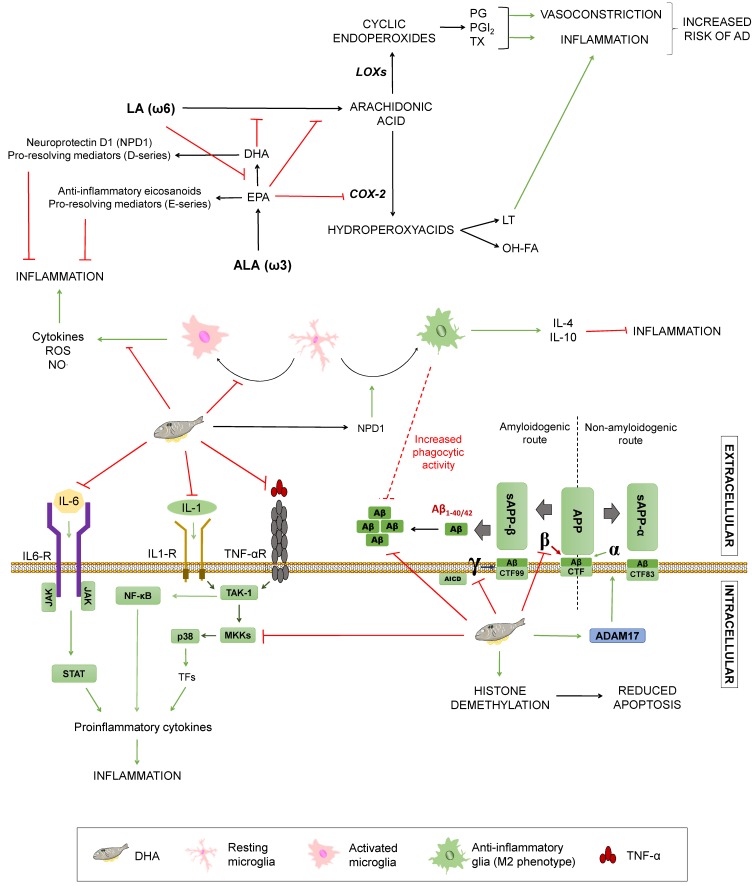
Effects of unsaturated fats on Alzheimer’s disease. Summary of the processes by which unsaturated fats may modulate Alzheimer’s Disease. COX-2: Ciclooxygenase-2. LOXs: Lipoxygenases. LT: Leukotrienes. OH-FA: hydroxy-fatty acids. PG: prostaglandins. PGI2: prostacyclin. TX: thromboxanes. NO: nitric oxide. TNFα: Tumour Necrosis Factor α. TNF-αR: TNFα receptor. IL-1: interleukin-1. IL-1R: IL-1 receptor. IL-6: interleukin-6. TAK1: Transforming growth factor beta-activated kinase 1. NF-κB: nuclear factor kappa-light-chain-enhancer of activated B cells. P38: P38 mitogen-activated protein kinases. MKKs: Mitogen-Activated Protein Kinase Kinases. STAT: signal transducer and activator of transcription. APP: amyloid-beta precursor protein. sAPPα: soluble α-APP. sAPPβ: soluble β-APP. ADAM17: ADAM Metallopeptidase Domain 17. CTF: carboxy-terminal fragment. AICD: APP intracellular domain. Green arrows represent activation. Red truncated arrows represent inhibition.

**Figure 5 diseases-07-00012-f005:**
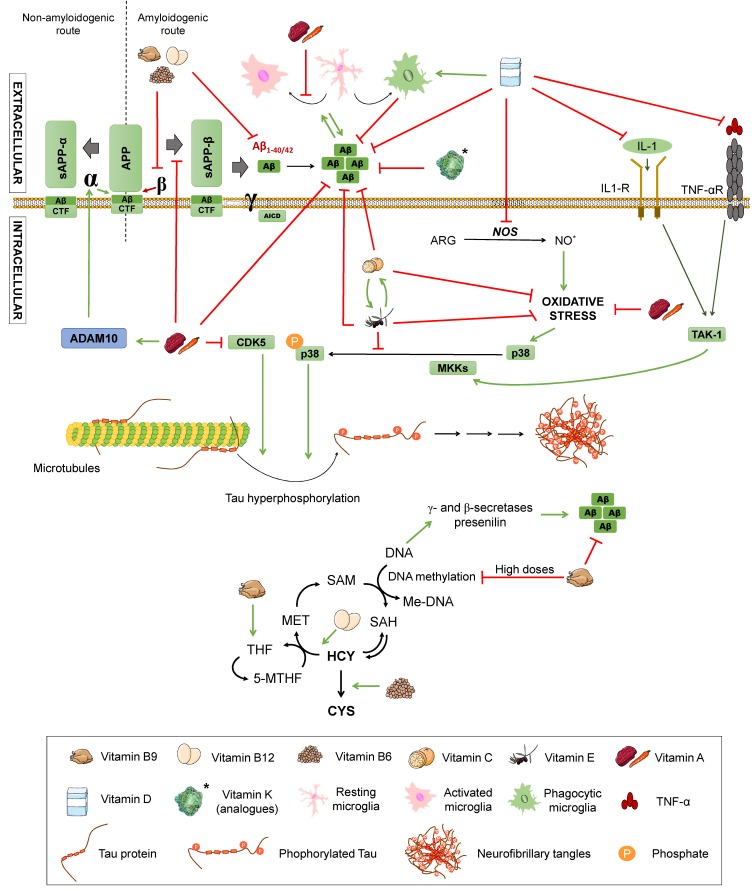
Effects of vitamins on Alzheimer’s disease. Effect of different vitamins on Aβ and Tau pathology, inflammation, and oxidative stress. ARG: arginine. NO: nitric oxide. TNFα: Tumour Necrosis Factor α. TNF-αR: TNFα receptor. IL-1: interleukin-1. IL-1R: IL-1 receptor. TAK1: transforming growth factor beta-activated kinase 1. P38: P38 mitogen-activated protein kinases. MKK4: Mitogen-Activated Protein Kinase Kinase 4. CDK5: cyclin-dependent kinase 5. APP: amyloid-beta precursor protein. sAPPα: soluble α-APP. sAPPβ: soluble β-APP. ADAM10: ADAM Metallopeptidase Domain 10. CTF: carboxy-terminal fragment. AICD: APP intracellular domain. Homocysteine metabolic pathway. MET: methionine. SAM: S-adenosylmethionine. SAH: S-adenosylhomocysteine. HCY: homocysteine. CYS: cysteine. THF: tetrahydrofolate. 5-MTHF: 5-methyltetrahydrofolate. Met-DNA: methylated-DNA. Green arrows represent activation. Red truncated arrows represent inhibition.

**Figure 6 diseases-07-00012-f006:**
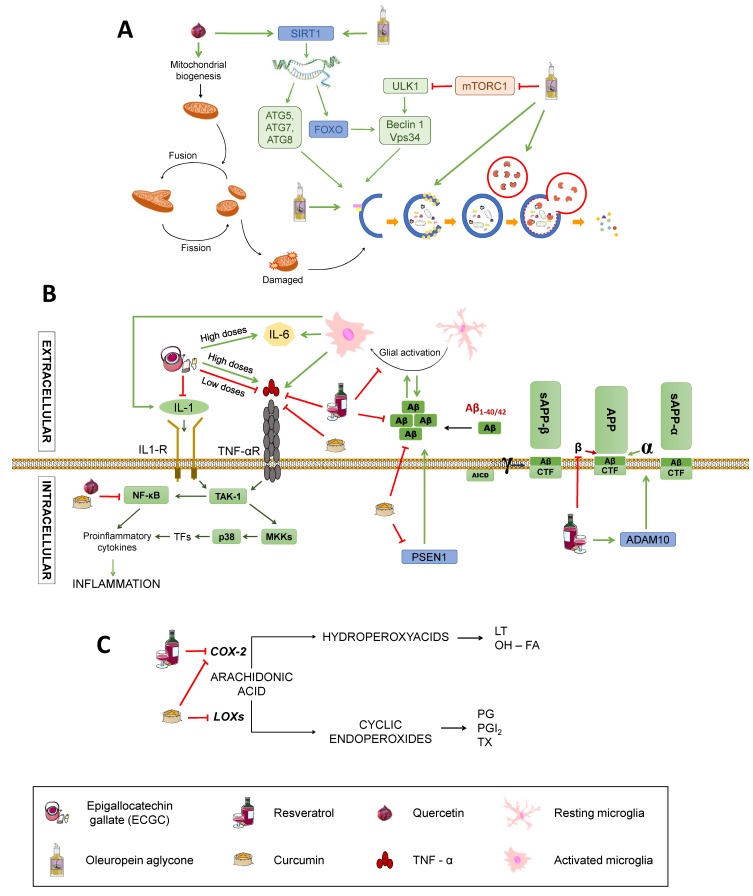
Effects on inflammation signalling pathways and Aβ production; (**C**) Effects on proinflammatory mediators’ synthesis. SIRT1: Sirtuin1. ATG: Autophagy-related protein. ULK1: unc-51 like autophagy activating kinase 1. FOXO: Forkhead box class “O” proteins. VPS34: Vacuolar protein sorting 34. mTORC1: mammalian target of rapamycin complex 1; TNFα: Tumour Necrosis Factor α. TNF-αR: TNFα receptor. IL-1: interleukin-1. IL-1R: IL-1 receptor. IL-6: interleukin-6. TAK1: transforming growth factor beta-activated kinase 1. NF-κB: nuclear factor kappa-light-chain-enhancer of activated B cells. P38: P38 mitogen-activated protein kinases. MKK4: Mitogen-Activated Protein Kinase Kinase 4. CDK5: cyclin-dependent kinase 5. APP: amyloid-beta precursor protein. sAPPα: soluble α-APP. sAPPβ: soluble β-APP. ADAM10: ADAM Metallopeptidase Domain 10. PSEN1: Presenilin-1. CTF: carboxy-terminal fragment. AICD: APP intracellular domain. (**C**) Effects on inflammation. COX-2: Ciclooxygenase-2. LOXs: Lipoxygenases. LT: Leukotrienes. OH-FA: hydroxy-fatty acids. PG: prostaglandins. PGI_2_: prostacyclin. TX: thromboxanes. Green arrows represent activation. Red truncated arrows represent inhibition.

**Figure 7 diseases-07-00012-f007:**
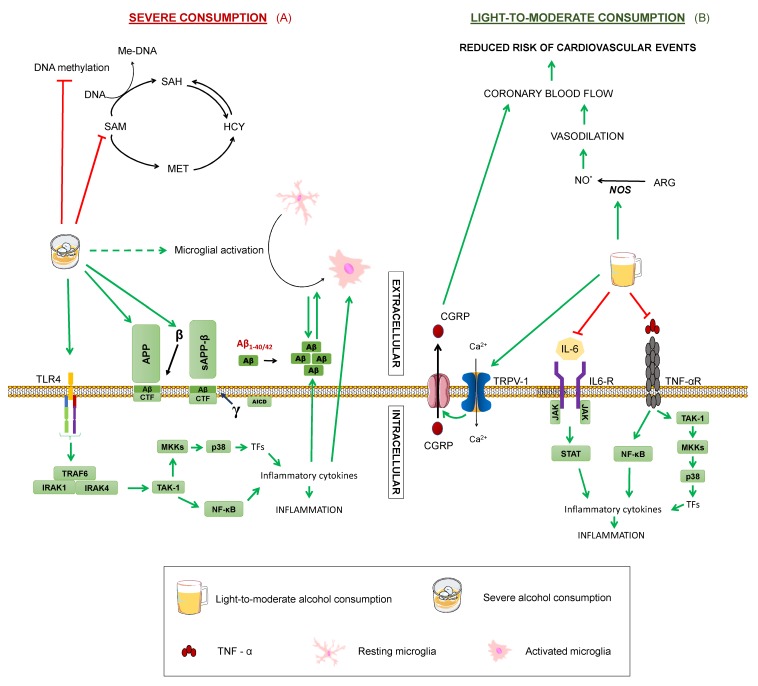
Effects of alcohol on Alzheimer’s disease. Summary of the processes by which alcohol may modulate Alzheimer’s disease in (**A**) severe consumption and (**B**) light-to-moderate: Tumour Necrosis Factor α. TNF-αR: TNFα receptor. IL-6: interleukin-6. TAK1: transforming growth factor beta-activated kinase 1. NF-κB: nuclear factor kappa-light-chain-enhancer of activated B cells. P38: P38 mitogen-activated protein kinases. MKK4: Mitogen-Activated Protein Kinase Kinase 4. TRAF6: TNF receptor associated factor 6. IRAK1: Interleukin-1 receptor-associated kinase 1. IRAK4: Interleukin-1 receptor-associated kinase 4. APP: amyloid-beta precursor protein. sAPPα: soluble α-APP. sAPPβ: soluble β-APP. CTF: carboxy-terminal fragment. AICD: APP intracellular domain. MET: methionine. SAM: S-adenosylmethionine. SAH: S-adenosylhomocysteine. HCY: homocysteine. THF: tetrahydrofolate. 5-MTHF: 5-methyltetrahydrofolate. Me-DNA: methylated-DNA. ARG: arginine. NO: nitric oxide. TRPV-1: transient receptor potential cation channel family V (vanilloid) member 1. CGRP: Calcitonin gene-related peptide. TNFα Green arrows represent activation. Dashed green arrows show indirect activation. Red truncated arrows represent inhibition.

**Figure 8 diseases-07-00012-f008:**
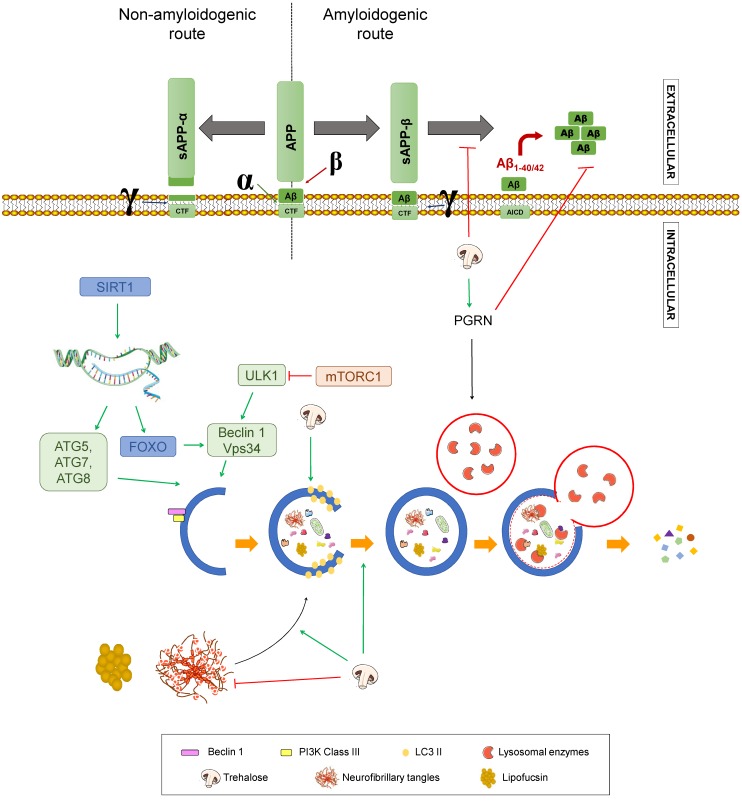
Effect of trehalose on Alzheimer’s disease. Summary of the processes by which trehalose may modulate Alzheimer’s disease. APP: amyloid-beta precursor protein. sAPPα: soluble α-APP. sAPPβ: soluble β-APP. CTF: carboxy-terminal fragment. AICD: APP intracellular domain. SIRT1: Sirtuin 1. ATG: Autophagy-related gen. ULK1: unc-51 like autophagy activating kinase 1. FOXO: Forkhead box class “O” proteins. VPS34: Vacuolar protein sorting 34. mTORC1: mammalian target of rapamycin complex 1. Green arrows represent activation. Red truncated arrows represent inhibition.

**Table 1 diseases-07-00012-t001:** Nutrient-rich food products for those nutrients that may intervene in Alzheimer’s disease.

Nutrients	Food Products			
DHA	Oily Fish *Salmon*	Nuts *Walnuts, Almonds*		egg yolk
Vitamin A	Meat *Chicken, turkey, beef*	Vegetables *Carrot, broccoli, sweet potato, pumpkin, spinach*	Dairy *Milk, cheese*	Fruit *Melon, apricot, papaya*
Vitamins C and E	Swiss chard	Cranberries and currants		Black olives
Vitamin B12	Meat *Pork, beef liver*	Fish *Salmon, sardines, mackerel, clams*		Egg
Vitamin B9	Meat *Chicken, beef*	Vegetables and legumes *Spinach, lentils, soy*		Egg yolk
Vitamin B6	Meat *Chicken, turkey, beef, pork*	Vegetables and fruits *Spinach, green peas, broccoli, asparagus, banana*	Fish *Salmon*	Nuts *Sunflower seeds, hazelnut, cashew nut*
EGCG	Green tea	Blackberry		Apple
Quercetin	Capers	Dark chocolate	Red onion	Clove
Resveratrol	Red wine	Berries *Blackberry, raspberry, currant*	Dark chocolate	Nuts *Walnuts, peanuts*
Oleuropein aglycone	Extra-virgin olive oil			
Trehalose	Mushrooms			

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
