# Peer review of "Modulating Effect of Diet on Alzheimer’s Disease"

_diseases, 2019, doi:10.3390/diseases7010012_

Round 1
Reviewer 1 Report
In this revised version, the authors extensively modified the organization and the content of their review. Although they wanted to keep the examination of the contributions of various nutrients rather than focusing on a few as I had suggested in my previous review, they succeeded to present a cohesive overview of the nutrition factors in Alzheimer’s disease (AD). Moreover, the authors underline that daily diet is composed of various nutrients which can have complementary but also opposite effects and this has to be taken into account in putative preventive strategies against AD. As I requested in my previous review, they clearly separated data obtained in animal or cellular models from those obtained in humans and they added several comments indicating the complexity of the analysis of the data obtained in clinical studies. Therefore, I consider that this review can be published after minor revisions:
- In Figure 1 and the above comment, the authors indicate that “although it is worth noting that, under pathological circumstances, β-secretase cleavage can ensue α-secretase, thus producing the truncated peptide Aβ1-16”. To my knowledge α- and β-secretases do not act in the same cell compartment. If Aβ1-16 peptide is produced in such pathway, it should be a minor component in the pathological AD mechanism. The authors should precise what they mean by “pathological circumstances”, indicate the importance of this catabolic pathway relatively to the whole effects of Aβ1-42 peptide and add at least a reference on this aspect.
- Since several studies indicate that obesity and type 2 diabetes (in which oxidative stress and inflammation are involved) contribute to the dementia risk and the control of these public health concerns could lead to the reduction of the senile dementia and AD prevalence, I would suggest to the authors to add a comment of the nutrition-based preventions of obesity and diabetes and the consecutive consequences on AD occurrence. This comment could take place for example in conclusion to avoid excessive lengthening of the review.
Author Response
Response to Reviewer 1 Comments
Point 1: In Figure 1 and the above comment, the authors indicate that “although it is worth noting that, under pathological circumstances, β-secretase cleavage can ensue α-secretase, thus producing the truncated peptide Aβ1-16”. To my knowledge α- and β-secretases do not act in the same cell compartment. If Aβ1-16 peptide is produced in such pathway, it should be a minor component in the pathological AD mechanism. The authors should precise what they mean by “pathological circumstances”, indicate the importance of this catabolic pathway relatively to the whole effects of Aβ1-42 peptide and add at least a reference on this aspect.
Response 1: This paragraph has been rewritten as follows: ”However, it is worth noting that is has been described that β-secretase cleavage can ensue α-secretase, thus producing alternative shorter non-aggregative Aβ fragments [14,19]. Among them, truncated peptide Aβ1 – 16 has been found to be increased in the CSF of both familial and sporadic AD patients together with a decrease in Aβ1 – 42, suggesting a possible contribution to the disease [20,21] (Figure 1).” Four relevant references have been included to clarify reviewer’s concerns. These articles discuss the subcellular location of α- and β-secretases and the potential circumstances in which this alternative cleavage takes place, to which extent and its relevance for the disease. As this information is not fully explained so far, we have not dwelt on these issues, which are far from the main interest of our article. By “pathological circumstances”, we were referring to the fact that this cleavage is supposed to take place in familial and sporadic Alzheimer’s disease because Aβ1 – 16 is increased in the CSF of these patients. To be more precise “pathological situation” and “non-pathological situation” labels have been removed from the figure and the corresponding sentence in the text was rewritten more accurately as we mentioned.
Point 2: Since several studies indicate that obesity and type 2 diabetes (in which oxidative stress and inflammation are involved) contribute to the dementia risk and the control of these public health concerns could lead to the reduction of the senile dementia and AD prevalence, I would suggest to the authors to add a comment of the nutrition-based preventions of obesity and diabetes and the consecutive consequences on AD occurrence. This comment could take place for example in conclusion to avoid excessive lengthening of the review.
Response 2: We agree with the reviewer about the relevance of obesity and diabetes as a dementia risk; therefore, we have included the following paragraph with the corresponding references in the text: “In addition, some dietary patterns may affect AD risk and development indirectly, through modulation of some AD risk factors. For instance, obesity, type 2 diabetes or cardiovascular disease have been established to increase the risk of developing AD [8,15,39,50] and certain dietary and lifestyle interventions have been proven to prevent or alleviate the impact of these conditions in health. Thus, this improvement may translate as a diminished risk of developing AD or other type of dementia [51].”
Relevant references for reviewer 1 comments:
8. Arizaga, R.; Barreto, D.; Bavec, C.; Berríos, W.; Cristalli, D.; Colli, L.; Garau, M.L.; Golimstok, A.; Ollari, J.; Sarasola, D. Dieta y prevención en enfermedad de alzheimer. Neurología Argentina 2018, 10, 44-60.
14. Cheignon, C.; Tomas, M.; Bonnefont-Rousselot, D.; Faller, P.; Hureau, C.; Collin, F. Oxidative stress and the amyloid beta peptide in alzheimer’s disease. Redox biology 2017.
15. González Rodríguez, L.G.; Palmeros Exsome, C.; González Martínez, M.T.; Pérez Ávila, M.d.l.L.; Gutiérrez López, M. Factores dietéticos y nutricionales en la prevención de la enfermedad de alzheimer. Revista Salud Pública y Nutrición 2016, 15, 27-37.
19. Portelius, E.; Price, E.; Brinkmalm, G.; Stiteler, M.; Olsson, M.; Persson, R.; Westman-Brinkmalm, A.; Zetterberg, H.; Simon, A.J.; Blennow, K. A novel pathway for amyloid precursor protein processing. Neurobiology of aging 2011, 32, 1090-1098.
20. Portelius, E.; Mattsson, N.; Andreasson, U.; Blennow, K.; Zetterberg, H. Novel aβisoforms in alzheimer's disease-their role in diagnosis and treatment. Current pharmaceutical design 2011, 17, 2594-2602.
21. Portelius, E.; Zetterberg, H.; Andreasson, U.; Brinkmalm, G.; Andreasen, N.; Wallin, A.; Westman-Brinkmalm, A.; Blennow, K. An alzheimer's disease-specific β-amyloid fragment signature in cerebrospinal fluid. Neuroscience letters 2006, 409, 215-219.
39. Heneka, M.T.; Carson, M.J.; El Khoury, J.; Landreth, G.E.; Brosseron, F.; Feinstein, D.L.; Jacobs, A.H.; Wyss-Coray, T.; Vitorica, J.; Ransohoff, R.M. Neuroinflammation in alzheimer's disease. The Lancet Neurology 2015, 14, 388-405.
50. Shinohara, M.; Sato, N. Bidirectional interactions between diabetes and alzheimer's disease. Neurochemistry international 2017, 108, 296-302.
51. Medina-Remón, A.; Kirwan, R.; Lamuela-Raventós, R.M.; Estruch, R. Dietary patterns and the risk of obesity, type 2 diabetes mellitus, cardiovascular diseases, asthma, and neurodegenerative diseases. Critical reviews in food science and nutrition 2018, 58, 262-296.
Reviewer 2 Report
The authors resubmitted a new manuscript regarding the potential beneficial role of (Mediterranean) diet on the incidence and progression of Alzheimer’s disease.
Despite the comments of both reviewers regarding the extent of the review on the mechanistic insights and the need to focus on some (micro)nutrients, the authors did use their previous work as a template for the resubmitted elaborated work. Therefore, they decided to spend again several pages on the main mechanism, as well as on many aspects of the disease. The relevant submitted figures are representative of the diverging and fragmented studies that have been performed over the years, and could be useful to the reader to follow literature.
The second point that both reviewers had addressed was that despite so many studies in vitro or the in vivo (models) showed interference of so many dietary compounds with molecular mechanisms, many interventions have failed to replicate the findings. The gap among bench studies and translational/clinical research is evident. I still believe that although it is mentioned now sporadically, the authors failed to reorganize their manuscript in a way to recognize it. This is the most common trait among reviews in the field, and seriously misleading to the increasing population of dementia sufferers.
The authors highlight the importance of the adoption of a diet, when a subject has been diagnosed or when a subject is at higher risk. Both considerations are weak, as i) the role of any diet has been shown contributing to neurodegenerative disorders, while the official point of WHO, as well as of (dietician) bodies regarding adequacy, thus diet content, are based on evidence linked to neurodegeneration either directly or indirectly. Moreover, the only established subjects at risk are those with family history of the disease, as well as ApoEε4 carriers, just a small fraction of people suffering from AD.
An essential contribution to the field would be to describe specific requirements, above and beyond DRIs, or to describe DRIs for unspecified yet categories. Even if these fall beyond the capacity or priority of the authors, any category should be at least discussed in view of the international dietary standards and suggested supplementations. This is definitely not a systematic review, where specific numeric data could be retrieved, however, it does lack such a perspective.
Quoting authors “The present review gathers the scientific evidence existing up‐to-date showing the modulating effect of diet on Alzheimer’s disease, due to the actions of nutrients on different processes, such as Aβ accumulation, tau hyperphosphorylation, oxidative stress, inflammation or autophagy… enlightening once again the importance of a proper diet in health maintenance.” The current work, still in its new format, gathers evidence of isolated compounds and several diets, but does not provide concrete information about the in vivo benefit of specific dietary components. Finally, what is a “proper diet”?
Author Response
Response to Reviewer 2 Comments
Point 1: Despite the comments of both reviewers regarding the extent of the review on the mechanistic insights and the need to focus on some (micro)nutrients, the authors did use their previous work as a template for the resubmitted elaborated work. Therefore, they decided to spend again several pages on the main mechanism, as well as on many aspects of the disease. The relevant submitted figures are representative of the diverging and fragmented studies that have been performed over the years, and could be useful to the reader to follow literature.
Response 1: As we previously argued, the aim of the review was to give a general wide view of the relevance of diet for Alzheimer’s disease in order to propose dietary intervention as a relevant factor to take into consideration for these patients. Another work focusing deeply into the specific mechanisms of some micronutrients would be undoubtedly very interesting but is far from the purpose of our current work.
Point 2: The second point that both reviewers had addressed was that despite so many studies in vitro or the in vivo (models) showed interference of so many dietary compounds with molecular mechanisms, many interventions have failed to replicate the findings. The gap among bench studies and translational/clinical research is evident. I still believe that although it is mentioned now sporadically, the authors failed to reorganize their manuscript in a way to recognize it. This is the most common trait among reviews in the field, and seriously misleading to the increasing population of dementia sufferers.
Response 2: We completely agree with the reviewer in the limitations of the studies performed so far and that there is an enormous gap between bench and bed studies. In fact, our work tried to reinforce the idea that more studies towards this direction are mandatory in order to decipher which would be the dietary patterns that these patients should follow. To strengthen this point, we have rewritten several parts of the article:
· In the abstract we have included the following sentence: “However, the extent to which these effects come with beneficial clinical outcomes remains unclear.”
· We have rewritten the last sentence of the abstract as: “These indications highlight the potential role of adequate dietary recommendations for clinical management of both Alzheimer’s diagnosed patients and those in risk of developing it, emphasising once again the importance of diet on health.”
· In the section “Relevance of dietary patterns” we have adapted the last sentences of the first two paragraphs as: “which may be potentially helpful considering the lack of an effective treatment for this disease.” and “helping to prevent it in populations at risk or even slowing down its progression in the most optimistic scenario.”
· In the third paragraph of “Relevance of dietary patterns”, we have also included the sentence: “Although it is important to emphasise that there is a notable gap in scientific knowledge when it comes to consider dietary intervention on clinical research”
· We have concluded the section “Relevance of dietary patterns” with the following fragment: “Furthermore, we include information about studies conducted in humans regarding the discussed nutrients, but it is necessary to underline that, overall, clinical research has failed to prove clear preventive or therapeutic effects of any dietary intervention for Alzheimer’s disease. This is partly due to inherent problems in translational research and partly because of the notable number of confounding factors that may exist when evaluating a dietary intervention. However, the potential utility of dietary interventions drawn from the mentioned studies [43-45,51] constitutes a compelling argument in favour of research focusing on dietary patterns and the mechanisms by which combined nutrients may modulate chronic diseases as AD.”
· The conclusion section has been rewritten as: “[…] Altogether, this evidence brings to light the relevant role that nutritional intervention might play in these patients and in the population at risk of developing AD, enlightening once again the importance of dietary patterns in health maintenance. Further research is needed in order to clarify whether early intake of food containing these nutrients or, more likely, adherence to a certain dietary pattern including such foods, would be useful as a preventive method for AD. It is clear that nutrients have a direct or indirect effect on the processes that lead to the neurodegeneration observed in AD, but the extent to which these effects may translate into actual modulation of the disease remains unclear. Thus, it is crucial to analyse the effects of dietary intervention on the long term in these populations, to which epidemiologic longitudinal studies should be carried out, especially considering that AD starts years before the first symptoms appear. […]”
Point 3: The authors highlight the importance of the adoption of a diet, when a subject has been diagnosed or when a subject is at higher risk. Both considerations are weak, as i) the role of any diet has been shown contributing to neurodegenerative disorders, while the official point of WHO, as well as of (dietician) bodies regarding adequacy, thus diet content, are based on evidence linked to neurodegeneration either directly or indirectly. Moreover, the only established subjects at risk are those with family history of the disease, as well as ApoEε4 carriers, just a small fraction of people suffering from AD.
Response 3: The “importance of the adoption of a diet” has been adapted as we previously mentioned in point 2. With respect to the second point about “people at higher risk” we agree with the reviewer that there is a small proportion of patients whose susceptibility can be genetically traced. However, as the majority of the patients are considered sporadic and the main risk factor is age, all population above 65 is considered population at risk (Karlawish, J.; Jack Jr, C.R.; Rocca, W.A.; Snyder, H.M.; Carrillo, M.C. Alzheimer's disease: The next frontier—special report 2017. Alzheimer's & Dementia 2017, 13, 374-380). Dietary recommendations may be potentially beneficial for all of them, which in the context of a progressively ageing population, constitutes a non-negligible part of the population.
Importantly, health systems would be unlikely to incur meaningful costs from the implementation of such measures, in spite of being a significative portion of the population, since it would not require running lab tests or administrating dietary supplements, but only the labour of nutrition professionals. Supporting this argument, procedures involving screening of nutritional status and nutritional supplementation, which would be considerably more expensive, are already being applied. For instance, patients with high tHcy are offered vitamin B supplementation in Sweden and a similar model has been proposed to be highly cost-effective in the UK (see: Tsiachristas, A., & Smith, A. D. (2016). B-vitamins are potentially a cost-effective population health strategy to tackle dementia: Too good to be true?. Alzheimer's & Dementia: Translational Research & Clinical Interventions, 2(3), 156-161 and Smith, A. D., Refsum, H., Bottiglieri, T., Fenech, M., Hooshmand, B., McCaddon, A., ... & Obeid, R. (2018). Homocysteine and dementia: an international consensus statement. Journal of Alzheimer's Disease, 62(2), 561-570).
In addition, dietary recommendations might be beneficial for this populational segment as a whole, not only for AD, but also for prevention, alleviation or treatment of other age-related diseases (cardiovascular disease, for instance) and to avoid nutritional deficiencies, quite common in the elderly.
Point 4: An essential contribution to the field would be to describe specific requirements, above and beyond DRIs, or to describe DRIs for unspecified yet categories. Even if these fall beyond the capacity or priority of the authors, any category should be at least discussed in view of the international dietary standards and suggested supplementations. This is definitely not a systematic review, where specific numeric data could be retrieved, however, it does lack such a perspective.
Response 4: We fully agree with the reviewer in the fact that it is necessary to describe specific requirements, above and beyond DRIs. To reinforce this idea we have include this item in the conclusions in the following sentence: “Additionally, it is essential to focus future research on estimating the amount of each nutrient that is required to exert the beneficial action, especially referring these to Dietary Reference Intakes, along with information about toxic doses, food-drug interactions and ideal synergistic effects arising from defined combinations that may result in a diet specifically optimised for AD.”
The discussion in view of the international dietary standards and suggested supplementations would be a very interesting point, but it constitutes a complex issue since it would require comparing supplementation of single nutrients or foods to dietary patterns, which are not equivalent and much more studies should be considered if we want to address this question properly, excessively enlengthening our review. Moreover, it would fall beyond the purpose of our work, since we intended to highlight the molecular mechanisms by which nutrients may modulate different processes contributing to Alzheimer’s disease and link them to potentially beneficial dietary patterns.
Nonetheless, a secondary aim of this work is to attract scientific attention to the necessity of further research taking into account this and other limitations. All the modifications highlighted in point 2 have been made in order to reinforce this idea.
Point 5: Quoting authors “The present review gathers the scientific evidence existing up‐to-date showing the modulating effect of diet on Alzheimer’s disease, due to the actions of nutrients on different processes, such as Aβ accumulation, tau hyperphosphorylation, oxidative stress, inflammation or autophagy… enlightening once again the importance of a proper diet in health maintenance.” The current work, still in its new format, gathers evidence of isolated compounds and several diets, but does not provide concrete information about the in vivo benefit of specific dietary components. Finally, what is a “proper diet”?
Response 5: This is a review article; therefore, it cannot go further than the current state of the art of the question. We cannot provide concrete information about the in vivo benefit of specific dietary components because this is a pending issue in the field and also a difficult question to answer. Studies considering isolated nutrients or foods, nutrient supplementation or adherence to certain dietary patterns have been discussed in our review, but as we mention on the article, the potential beneficial effects are unlikely to arise from single dietary components but rather from the interaction between them in dietary patterns, and there are no studies to our knowledge that univocally link certain nutrient or food to a beneficial outcome. All these limitations have been described throughout the text, emphasised in the “Relevance of dietary patterns” section, including the modifications highlighted in response 2 and reprised in the conclusion (also mentioned in response 2).
With regards to the quoted sentence, we want to acknowledge that it may sound as quite a certain statement and that the term “proper diet” was ill-used. We have rewritten this sentence as follows: “The present review gathers the scientific evidence existing up-to-date showing the potential modulating effect of diet on Alzheimer’s disease, due to the actions of nutrients on different processes, such as Aβ accumulation, tau hyperphosphorylation, oxidative stress, inflammation or autophagy. […] enlightening once again the importance of dietary patterns in health maintenance.”
Relevant references for reviewer 2 comments:
43. Singh, B.; Parsaik, A.K.; Mielke, M.M.; Erwin, P.J.; Knopman, D.S.; Petersen, R.C.; Roberts, R.O. Association of mediterranean diet with mild cognitive impairment and alzheimer's disease: A systematic review and meta-analysis. Journal of Alzheimer's disease 2014, 39, 271-282.
44. Petersson, S.D.; Philippou, E. Mediterranean diet, cognitive function, and dementia: A systematic review of the evidence–. Advances in Nutrition 2016, 7, 889-904.
45. Martínez-Lapiscina, E.H.; Clavero, P.; Toledo, E.; Estruch, R.; Salas-Salvadó, J.; San Julián, B.; Sanchez-Tainta, A.; Ros, E.; Valls-Pedret, C.; Martinez-Gonzalez, M.Á. Mediterranean diet improves cognition: The predimed-navarra randomised trial. J Neurol Neurosurg Psychiatry 2013, jnnp-2012-304792.
51. Medina-Remón, A.; Kirwan, R.; Lamuela-Raventós, R.M.; Estruch, R. Dietary patterns and the risk of obesity, type 2 diabetes mellitus, cardiovascular diseases, asthma, and neurodegenerative diseases. Critical reviews in food science and nutrition 2018, 58, 262-296.
Reviewer 3 Report
The manuscript i interesting a suitable for the journal
Author Response
Thank you very much for your interest and consideration
Round 2
Reviewer 2 Report
At this point, the authors made corrections to support their choice towards the presentation of the relevance of dietary patterns, and individual dietary components, for the incidence and progression of AD. It is still arguable whether the acclaimed benefits derive after meeting or exceeding the current recommended values, as well as whether there is synergy among dietary components. As initially suggested, the authors had integrated the recent consensus for homocysteine levels. Although not relevant to a Mediterrannean pattern, not adequate for high-quality protein and substantial B12 intake in the elderly, the information has been used to explain recent advances and diagnostic approaches. Such standard procedures are imperative during any dietary evaluation. The claims about the high cost of laboratory analysis had been detrimental so far, as assumptions about dietary intake were made, mostly based on previous population studies. This is definitely not the current picture. Despite higher energy in the modern world, nutrient deficiency is extremely high, e.g. vitamin E (detected through blood tests) even in olive-oil producing countries! Therefore, it would be essential to integrate all relevant information about "supplementation" or dietary intake (range, frequency, type, comparison to DRIs) from citations, and provide scientifically relevant information to the reader.